# The Rank-Reduced Kalman Filter: Approximate Dynamical-Low-Rank Filtering In High Dimensions

**Jonathan Schmidt**
Tübingen AI Center, University of Tübingen
`jonathan.schmidt@uni-tuebingen.de`

**Philipp Hennig**
Tübingen AI Center, University of Tübingen
`philipp.hennig@uni-tuebingen.de`

**Jörg Nick**
University of Tübingen
`nick@na.uni-tuebingen.de`

**Filip Tronarp**
Lund University
`filip.tronarp@matstat.lu.se`

## Abstract

Inference and simulation in the context of high-dimensional dynamical systems remain computationally challenging problems. Some form of dimensionality reduction is required to make the problem tractable in general. In this paper, we propose a novel approximate Gaussian filtering and smoothing method which propagates low-rank approximations of the covariance matrices. This is accomplished by projecting the Lyapunov equations associated with the prediction step to a manifold of low-rank matrices, which are then solved by a recently developed, numerically stable, dynamical low-rank integrator. Meanwhile, the update steps are made tractable by noting that the covariance update only transforms the column space of the covariance matrix, which is low-rank by construction. The algorithm differentiates itself from existing ensemble-based approaches in that the low-rank approximations of the covariance matrices are deterministic, rather than stochastic. Crucially, this enables the method to reproduce the exact Kalman filter as the low-rank dimension approaches the true dimensionality of the problem. Our method reduces computational complexity from cubic (for the Kalman filter) to *quadratic* in the state-space size in the worst-case, and can achieve *linear* complexity if the state-space model satisfies certain criteria. Through a set of experiments in classical data-assimilation and spatio-temporal regression, we show that the proposed method consistently outperforms the ensemble-based methods in terms of error in the mean and covariance with respect to the exact Kalman filter. This comes at no additional cost in terms of asymptotic computational complexity.

## 1 Introduction

Spatio-temporal dynamical systems have always played an important role in the applied sciences, such as climate science, numerical weather prediction, or geophysics. In precisely these settings, one is often faced with massive amounts of data to be processed. At the same time the interactions are of such complex nature that it is imperative to include a notion of uncertainty in the model and in its outputs. Both these requirements have a reputation as being computationally expensive. Hence, sensible approximations are indispensable at a certain scale.

Concretely, we consider state-space models of the form

$$\mathrm{dx}(t) = \mathrm{Ax}(t)\,\mathrm{d}t + \mathrm{B}\,\mathrm{dw}(t), \tag{1a}$$

$$\mathrm{y}(t_l) \mid \mathrm{x}(t_l) \sim \mathcal{N}(\mathrm{Cx}(t_l), \mathrm{R}), \quad l = 1, \dots, N \tag{1b}$$

37th Conference on Neural Information Processing Systems (NeurIPS 2023).

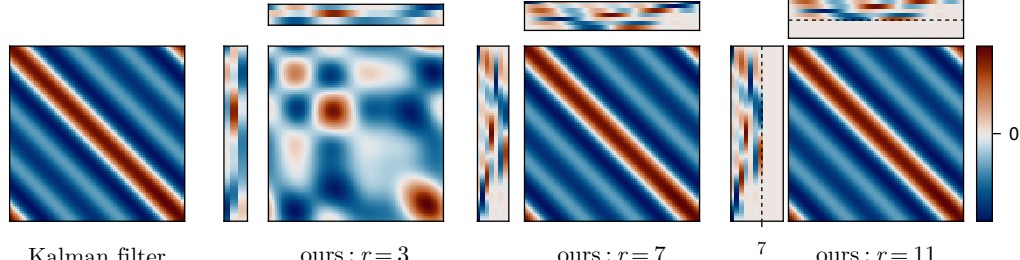

Figure 1: *Rank-$r$ approximations to the true KF covariance (left) for increasing $r$.* The considered problem's true rank is $r^* = 7$ by construction, the state-dimension is $n = 1000$. The respective low-rank factors are shown above and left of their outer product. For $r \geq r^*$, the KF estimate is recovered. For $r > r^*$, the excess columns of the low-rank factor collapse (rightmost plot).

where $\mathrm{x} \in \mathbb{R}^n$ is the latent state, $\mathrm{y} \in \mathbb{R}^m$ is the measurement process, and w is a standard Wiener process in $\mathbb{R}^q$. Equation (1a) defines the prior dynamics model via a linear time-invariant (LTI) stochastic differential equation (SDE) with drift matrix $\mathrm{A} \in \mathbb{R}^{n \times n}$ and dispersion matrix $\mathrm{B} \in \mathbb{R}^{n \times q}$. The linear Gaussian observation model Eq. (1b) is specified via the measurement matrix $\mathrm{C} \in \mathbb{R}^{m \times n}$ and measurement-noise covariance $\mathrm{R} \in \mathbb{R}^{m \times m}$, where possible time-dependency has been omitted from the notation. While the probabilistic state-estimation problem may in principle be solved in closed form using the Kalman filter (KF) and smoother [25, 34], we will consider the case where the state dimension $n$ and measurement dimension $m$ are very large, making the covariance recursion computationally prohibitive in practice. Such settings include spatio-temporal Gaussian process regression [35, 37, 45, 7, 20], and data assimilation applications such as meteorology and oceanography [17, 11], numerical weather forecasting [21], geoscience [6], inverse problems [9, 41], and brain imaging [16]. While the Gaussian process (GP) literature offers many ways of tractable, approximate inference, approaches based on approximations of the underlying kernel function are not amenable to the present problem setting. Without a kernel to approximate, the problem is often solved by ensemble methods, which stochastically propagate low-rank approximations to the covariances in the filtering recursion. This stochasticity introduces unfavorable properties in these methods. To avert this shortcoming, we propose a novel low-rank filtering recursion, which is fully deterministic.

**Contributions** This text develops a method for efficient, approximate Gaussian filtering, including smoothing and marginal likelihood computation. The method employs the standard predict/correct formulation of the filtering problem, and can in principle be thought of as a square-root implementation of a Kalman filter, in that it represents all covariance matrices as their left square-root factors [19]. However, instead of square matrix square roots, the covariances are approximated by $(n \times r)$-dimensional low-rank factors, where $r \ll n$. We propose an approximate low-rank prediction-correction loop, together with the computation of the backwards Markov process representing the smoothing posterior, all while preserving this low-rank structure throughout time.

Crucial to this work are (i) the use of dynamic-low-rank approximation (DLRA) [27, 29, 8] to solve Lyapunov equations on the manifold of rank-$r$ matrices in the prediction step and (ii) an efficient correction scheme that updates only the low-dimensional column space of the predicted covariance factor, which introduces no additional approximation error. The computational complexity of the proposed filter is asymptotically equal to those of the widely-used ensemble Kalman filters.

The method offers a deterministic alternative to existing ensemble methods. In particular, it recovers the exact Kalman filter estimate both in the full-rank limit $r = n$ and if the problem is in fact of rank $r < n$. The latter is demonstrated in Fig. 1, in which a problem with $n = 1000$ state dimensions and a true rank of 7 is approximated with $r = 3, 7, 11$. As soon as $r$ exceeds the true rank of the problem, the residual subspace dimensions collapse. The method will be detailed in Section 3 and evaluated on the basis of various experiments in Section 4. All source code is publicly available on GitHub.[1,2]

---

[1] https://github.com/schmidtjonathan/RRKF.jl
[2] https://github.com/schmidtjonathan/RRKF_experiments/

**Related work** The state estimation problem defined by Eq. (1) amounts to linear Gaussian state estimation and can be computed in closed form by Kalman filtering and smoothing. A modern overview can be found in Särkkä and Solin [36]. While the exact filtering and smoothing recursions scale linearly with the number of time points $N$, they scale cubically in the state dimension $n$ and the measurement dimension $m$, making their use computationally infeasible in many high-dimensional problems. A widely-used approximate filtering technique used to circumvent this computational burden are ensemble Kalman filters (EnKF) [11–13, 26, 32, 6]. The ensemble is a set of $r$ randomly sampled state vectors of dimension $n$, which parametrize the mean and covariance via sample statistics. It is processed successively in a prediction-correction loop similar to the KF. The mean-centered ensemble can be regarded as a low-rank square-root factor of the sample covariance matrix. In Section 4 we compare our method to the standard version of the EnKF and the ensemble transform Kalman filter [4], which is categorized as an ensemble-square-root filter [43].

In so far as it aims for computationally tractable Gaussian process regression, the proposed method is related to approaches using the Nyström method [48], inducing point methods [38, 44, 22], random Fourier features [31], and doubly-sparse variational Gaussian processes [1].

The literature offers more contributions related to the present work under the broader context of low-rank and dimensionality-reduction methods. Loiseau et al. [28], Netto and Mili [30], Vijayshankar et al. [47] approach low-rank modelling of dynamical systems based on measured data, whereas Farrell and Ioannou [15] are concerned with low-rank approximations of a given model based on classical theory on linear time-invariant systems. Cressie et al. [10] develop a tractable hierarchical Bayesian model to approximate high-dimensional spatio-temporal dynamics in a reduced latent space and condition on measurements by transporting the low-dimensional diffusive process to the full space by a projection that is assumed to be known. In those cases, once a low rank model is obtained for the phenomena under study, state estimation becomes computationally tractable, due to the reduced size of the resulting model. These works build upon the assumption that the state of the system evolves approximately in a low-dimensional manifold, whereas our method works under the assumption that the *error* in the state estimate evolves approximately in a low-dimensional manifold.

## 2 Background

Throughout this work, for a function $x(t)$ and a grid $(t_1, .., t_N)$ we use the abbreviated notation $x(t_l) = x_l$ and $(x(t_a), ...x(t_b)) = x_{a:b}$. The task of interest is computing the filtering densities $p(x_l \mid y_{1:l})$ and smoothing densities $p(x_l \mid y_{1:N})$ for $l = 1, \ldots, N$. LTI SDEs of the form in Eq. (1a) admit an equivalent discrete formulation in terms of linear Gaussian transition densities of the form

$$\mathrm{x}_l \mid \mathrm{x}_{l-1} \sim \mathcal{N}(\Phi_l \mathrm{x}_{l-1}, \mathrm{Q}_l). \tag{2}$$

The transition matrix $\Phi(t) \in \mathbb{R}^{n \times n}$ and process-noise covariance matrix $\mathrm{Q}(t) \in \mathbb{R}^{n \times n}$ solve the matrix differential equations $\dot{\Phi}(t) = A\Phi(t)$ and

$$\dot{\mathrm{Q}}(t) = A\mathrm{Q}(t) + \mathrm{Q}(t)A^* + BB^*, \tag{3}$$

in the interval $[t_{l-1}, t_l)$ with initial conditions $\Phi(t_{l-1}) = I$ and $\mathrm{Q}(t_{l-1}) = 0$, respectively [36]. Equation (3) is known as a *Lyapunov equation*, and will be of special importance in this work.

### 2.1 Square-root filtering

For numerically stable filtering and smoothing, all covariance matrices involved can be represented by matrix square roots (e.g. Cholesky factors). This is known as square-root filtering/smoothing [19]. Let $(\mathrm{M}_1 \quad \mathrm{M}_2)$ denote a wide block matrix in $\mathbb{R}^{d_1 \times (d_2 + d_3)}$, where $\mathrm{M}_1 \in \mathbb{R}^{d_1 \times d_2}$ and $\mathrm{M}_2 \in \mathbb{R}^{d_1 \times d_3}$ are some matrices. Given the filtering covariance at time $t_{l-1}$ as $\Sigma_{l-1} = \Sigma_{l-1}^{1/2} \Sigma_{l-1}^{*/2}$ a square-root factorization of the predicted covariance $\Pi_l = \Pi_l^{1/2} \Pi_l^{*/2}$ is given by

$$\Pi_l = \Phi_l \Sigma_{l-1} \Phi_l^* + \mathrm{Q}_l = \left( \Phi_l \Sigma_{l-1}^{1/2} \quad \mathrm{Q}_l^{1/2} \right) \left( \Phi_l \Sigma_{l-1}^{1/2} \quad \mathrm{Q}_l^{1/2} \right)^*, \tag{4}$$

and, hence, $(\Phi_l \Sigma_{l-1}^{1/2} \quad \mathrm{Q}_l^{1/2})$ is a matrix square-root of $\Pi_l$. Similarly, the correction step can be carried out entirely on square-root factors. For more details, see Grewal and Andrews [19].

## 2.2 Dynamical low-rank approximation

Consider an initial value problem with a matrix-valued flow field $F : \mathbb{R} \times \mathbb{R}^{u \times v} \to \mathbb{R}^{u \times v}$

$$\dot{\mathrm{M}}(t) = F(t, \mathrm{M}(t)), \qquad \mathrm{M}(t_0) = \mathrm{M}_0, \tag{5}$$

where $u$ and $v$ are potentially very large. Dynamic low-rank approximation (DLRA) methods [27, 8] efficiently compute low-rank factorizations $\mathrm{Y}(t) = \mathrm{U}(t)\mathrm{D}(t)\mathrm{V}^*(t) \approx \mathrm{M}(t)$ of Eq. (5) by solving

$$\dot{\mathrm{Y}}(t) = \mathcal{P}_r[\mathrm{Y}(t)] \circ F(t, \mathrm{Y}(t)), \qquad \mathrm{Y}(t_0) = \mathrm{U}_0 \mathrm{D}_0 \mathrm{V}_0^*, \tag{6}$$

instead. The matrices $\mathrm{U}_0 \in \mathbb{R}^{u \times r}$, $\mathrm{D}_0 \in \mathbb{R}^{r \times r}$, and $\mathrm{V}_0 \in \mathbb{R}^{v \times r}$ are an initial low-rank factorization of $\mathrm{Y}(t_0) \approx \mathrm{M}(t_0)$. $\mathcal{P}_r[\mathrm{Y}(t)]$ denotes the projection operator onto the tangent space at $\mathrm{Y}(t)$, where $\mathrm{Y}(t)$ lies in the manifold of rank-$r$ matrices. In the present work, we leverage this technique to obtain a low-rank factor $\mathrm{Q}^{1/2}$ of the process-noise covariance at the next prediction location. Thus, we avoid much of the computational cost in the prediction step, otherwise caused by solving the full Lyapunov equation for $\mathrm{Q}$ in Eq. (3), for example using matrix-fraction decomposition [40, 3]. Concretely, let $F(\mathrm{Q}) = \mathrm{AQ} + \mathrm{QA}^* + \mathrm{BB}^*$. Then, an approximate low-rank process-noise covariance matrix associated with the prediction step is obtained by integrating

$$\dot{\mathrm{Q}}(t) = \mathcal{P}_r[\mathrm{Q}(t)] \circ F(\mathrm{Q}(t)), \qquad \mathrm{Q}(t_{l-1}) = \mathrm{U}_0 \mathrm{D}_0^2 \mathrm{U}_0^* = 0 \tag{7}$$

from $t_{l-1}$ to $t_l$. At the start of the filtering recursion $t_0$, an initial low-rank factorization is constructed with a random orthogonal matrix $\mathrm{U}_0$ and $\mathrm{D}_0 = 0$. For all subsequent steps the propagated orthogonal basis $\mathrm{U}(t_l)$ can be reused. In this work, all mentions of dynamic low-rank integration always refer to the recently developed, numerically stable basis update & Galerkin (BUG) integrator [8], whose error bounds are independent of small singular values. More details are given in Appendix C.

# 3 Rank-reduced Kalman filtering

In this section, a method for approximate inference in Eq. (1) is developed. The idea is to approximate the full filtering/smoothing covariance matrices of the latent state by an eigendecomposition truncated at the $r$-th largest eigenvalue. The main challenge is to thereby attain linear computational scaling in $n$ and $m$ under appropriate assumptions on the state-space model. Before proceeding with the filtering and smoothing recursions, it is instructive to examine inference in a static low-rank model.

## 3.1 Efficient inference in static models of low rank

Consider the following latent variable model

$$\mathrm{x} \sim \mathcal{N}(\mu, \Pi^{1/2}\Pi^{*/2}), \qquad \mathrm{y} \sim \mathcal{N}(\mathrm{Cx}, \mathrm{R}), \tag{8}$$

where $\Pi^{1/2} \in \mathbb{R}^{n \times r}$ is of full column rank, $r \leq n$, and $\mathrm{R} \in \mathbb{R}^{m \times m}$. Representing the latent state $\mathrm{x}$ as

$$\mathrm{x} = \mu + \Pi^{1/2}\mathrm{z}, \qquad \mathrm{z} \sim \mathcal{N}(0, \mathrm{I}_{r \times r}), \tag{9}$$

reduces the problem to inference in the following model

$$\mathrm{z} \sim \mathcal{N}(0, \mathrm{I}_{r \times r}), \qquad \mathrm{y} \mid \mathrm{z} \sim \mathcal{N}(\mathrm{C}\mu + \mathrm{C}\Pi^{1/2}\mathrm{z}, \mathrm{R}). \tag{10}$$

An efficient inference scheme is given by the following proposition, which is proved in Appendix A.1.

**Proposition 1.** *Let* $\mathrm{z}$ *and* $\mathrm{y}$ *be two random variables governed by Eq.* (10)*. Assume* $r \leq m$ *and consider the following singular value decomposition* $(\mathrm{R}^{-1/2}\mathrm{C}\Pi^{1/2})^* = \mathrm{UDV}^*$*, where* $\mathrm{U}, \mathrm{D} \in \mathbb{R}^{r \times r}$ *and* $\mathrm{V} \in \mathbb{R}^{m \times r}$*. Then, defining the whitened residual* $\mathrm{e} = \mathrm{R}^{-1/2}(\mathrm{y} - \mathrm{C}\mu)$*, we get*

$$\mathrm{y} \sim \mathcal{N}(\mathrm{C}\mu, \mathrm{R}^{1/2}(\mathrm{VD}^2\mathrm{V}^* + \mathrm{I})\mathrm{R}^{*/2}), \tag{11a}$$

$$\mathrm{z} \mid \mathrm{y} \sim \mathcal{N}(\mathrm{U}(\mathrm{I} + \mathrm{D}^2)^{-1}\mathrm{DV}^*\mathrm{e}, \mathrm{U}(\mathrm{I} + \mathrm{D}^2)^{-1}\mathrm{U}^*). \tag{11b}$$

*Furthermore, let* $|\cdot|$ *denote the matrix determinant. The marginal log-likelihood of* $\mathrm{y}$ *is given by*

$$\log \mathcal{N}(\mathrm{y}; \mathrm{C}\mu, \mathrm{R}^{1/2}(\mathrm{VD}^2\mathrm{V}^* + \mathrm{I})\mathrm{R}^{*/2}) = -\frac{m}{2}\log 2\pi - \log |\mathrm{R}^{1/2}| - \frac{1}{2}\sum_{k=1}^{r}\log(1 + \mathrm{D}_{kk}^2)$$
$$-\frac{1}{2}\|\mathrm{e}\|^2 + \frac{1}{2}\mathrm{e}^*\mathrm{VD}(\mathrm{D}^2 + \mathrm{I})^{-1}\mathrm{DV}^*\mathrm{e}. \tag{12}$$

The below corollary follows from the deterministic relationship between z and x given by Eq. (9).

**Corollary 1.** *Let* y *and* x *be two random variables governed by the model Eq.* (8) *and* $r \leq m$. *Then*

$$x \mid y \sim \mathcal{N}(\mu + Ke, \Sigma), \tag{13}$$

*where*

$$K = \Pi^{1/2}U(I + D^2)^{-1}DV^*, \qquad \Sigma = \Pi^{1/2}U(I + D^2)^{-1}U^*\Pi^{*/2}. \tag{14}$$

*Moreover, a square-root of* $\Sigma$ *is readily obtained by*

$$\Sigma^{1/2} = \Pi^{1/2}U(I + D^2)^{-1/2}. \tag{15}$$

## 3.2 The rank-reduced filtering recursion

In this section, a low-rank prediction-correction recursion is developed for the purpose of obtaining the filtering densities and the logarithm of the marginal likelihood.

**The prediction equations**    Suppose the filtering distribution at time $t_{l-1}$ is given by

$$p(x_{l-1} \mid y_{1:l-1}) = \mathcal{N}(x_{l-1}; \mu_{l-1}, \Sigma_{l-1}^{1/2}\Sigma_{l-1}^{*/2}), \tag{16}$$

where $\Sigma_{l-1}^{1/2} \in \mathbb{R}^{n \times r}$. We begin by detailing how to compute a low-rank factor $\Pi_l^{1/2}$ of the predicted covariance. First compute the truncated singular value decomposition (SVD)

$$\left(\Phi_l \Sigma_{l-1}^{1/2} \quad Q_l^{1/2}\right) \approx \tilde{U}_l \tilde{D}_l \tilde{V}_l^* \tag{17}$$

of the square-root factor (cf. Section 2.1), with $\tilde{U}_l \in \mathbb{R}^{n \times r}, \tilde{D}_l \in \mathbb{R}^{r \times r}$, and $\tilde{V}_l \in \mathbb{R}^{r \times r}$. This can be done in an optimal way by computing the full SVD and truncating it at the $r$-th largest singular value in $\mathcal{O}(nr^2)$ [18]. The process-noise covariance factor $Q_l^{1/2} \in \mathbb{R}^{n \times r}$ in Eq. (17) is computed using DLRA as described in Section 2.2. The rank-reduced predicted moments at time $t_l$ are thus given as

$$\mu_l^- = \Phi_l \mu_{l-1} \quad \in \mathbb{R}^n, \tag{18}$$
$$\Pi_l^{1/2} = \tilde{U}_l \tilde{D}_l \quad \in \mathbb{R}^{n \times r}. \tag{19}$$

**The update equations**    The low-rank update equations follow from Proposition 1 and Corollary 1:

$$\mu_l = \mu_l^- + K_l e_l \quad \in \mathbb{R}^n, \tag{20}$$
$$\Sigma_l^{1/2} = \Pi_l^{1/2}U_l(I + D_l^2)^{-1/2} \quad \in \mathbb{R}^{n \times r}, \tag{21}$$

with the whitened residual $e_l = R^{-1/2}(y_l - C\mu_l^-)$ and

$$(R^{-1/2}C\Pi_l^{1/2})^* = U_l D_l V_l^*, \qquad K_l = \Pi_l^{1/2}U_l(I + D_l^2)^{-1}D_l V_l^*, \tag{22}$$

for $U_l, D_l \in \mathbb{R}^{r \times r}$ and $V_l \in \mathbb{R}^{m \times r}$. The marginal predictive log-likelihood of $y_l$ is given by

$$\log p(y_l \mid y_{1:l-1}) = -\frac{m}{2}\log 2\pi - \log |R^{1/2}| - \frac{1}{2}\sum_{k=1}^{r}\log((D_l)_{kk}^2 + 1)$$
$$- \frac{\|e_l\|^2}{2} + \frac{1}{2}e_l^* V_l D_l (D_l^2 + I)^{-1}D_l V_l^* e_l. \tag{23}$$

Corollary 1—and by extension, the above filtering recursion—is only valid when $r \leq m$, which is exactly the intended usecase for the proposed method. For settings in which $m < r \leq n$ the correction step follows the square-root correction of a Kalman filter and is detailed in Appendix B.

## 3.3 Time complexity of the rank-reduced filtering recursion

This section analyzes the computational complexity of the proposed method. In the worst case, the method scales quadratically in the state dimension $n$ and measurement dimension $m$. Under favorable conditions, which are met in many real-world applications, the rank-reduced Kalman filter obtains a computational complexity of $O(nr^2)$ as stated by Proposition 2, which is proven in Appendix A.2. We begin by specifying a set of assumptions for Proposition 2.

**Assumption 1.** *The maps* $x \mapsto Ax$, $x \mapsto \Phi x$, *and* $x \mapsto BB^*x$ *can be evaluated in* $\mathcal{O}(n)$.

Assumption 1 is fulfilled naturally in many dynamical systems, in which local operators introduce sparsity into the dynamics, as, e.g., in finite-difference approximations of spatial differential operators. Another example is Kronecker structure in the system matrices arising in spatio-temporal GP regression by assuming the covariance function is separated into a product over the respective spatial and temporal components [35, 39, 20, 42].

**Assumption 2.** *The map* $x \mapsto Cx$ *can be evaluated in* $\mathcal{O}(m)$.

**Assumption 3.** *The map* $x \mapsto R^{-1/2}x$ *and the log-determinant* $\log |R^{1/2}|$ *can be evaluated in* $\mathcal{O}(m)$.

Throughout this work, we refer to the situation in which Assumption 1 or 2 do not apply as the "worst case". Assumption 3 is taken for granted as the measurement-noise covariance $R$ is often a diagonal matrix, which implies that sensor errors are uncorrelated. This is not only realistic but also commonly imposed in modelling. The above assumptions allow the following proposition.

**Proposition 2.** *Given Assumptions 1 to 3, the proposed method approximates the filtering densities and the marginal likelihood at a cost of* $\mathcal{O}(nr^2 + mr^2 + r^3)$.

The below corollary follows from the book-keeping in the proof of Proposition 2 in Appendix A.2.

**Corollary 2.** *When Assumption 1 or Assumption 2 are not satisfied, the worst-case complexity of the proposed low-rank filtering recursion is* $\mathcal{O}(n^2r + nm + m^2r)$.

Overall, the proposed filtering recursions achieve the same asymptotic complexity as the existing ensemble methods under the same set of assumptions. However, our method is entirely deterministic.

### 3.4 The rank-reduced smoothing recursion

It remains to obtain a recursion for the smoothing densities, which can be shown to be computationally tractable given that the filtering recursion is tractable. The smoothing densities are denoted by

$$p(x_l \mid y_{1:N}) = \mathcal{N}(x_l; \xi_l, \Lambda_l), \quad \xi_N = \mu_N, \ \Lambda_N = \Sigma_N. \tag{24}$$

It can be shown that the posterior process has a backward Markov representation [5], hence the smoothing marginals may be obtained by

$$p(x_l \mid y_{1:N}) = \int b_{l,l+1}(x_l \mid x_{l+1}) p(x_{l+1} \mid y_{1:N}) \, dx_{l+1}, \tag{25}$$

where we call $b_{l,l+1}$ the *backwards kernel*. Consequently, the problem consists of approximating $b_{l,l+1}$ such that the marginalization Eq. (25) may be implemented in a computationally frugal manner.

**Approximating the backwards kernel**  For the linear Gaussian case, the backward kernel [46]

$$b_{l,l+1}(x_l \mid x_{l+1}) = \mathcal{N}(x_l; G_l x_{l+1} + v_l, P_l), \tag{26}$$

is parametrized by the *smoothing gain* $G_l \in \mathbb{R}^{n \times n}$, the shift vector $v_l \in \mathbb{R}^n$, and the covariance of the backwards kernel $P_l \in \mathbb{R}^{n \times n}$. Let $(\cdot)^+$ denote the Moore–Penrose pseudoinverse. Then

$$G_l = \Sigma_l \Phi_{l+1}^* \Pi_{l+1}^+, \quad v_l = \mu_l - G_l \mu_{l+1}^-, \quad P_l = (I - G_l \Phi_{l+1}) \Sigma_l (I - G_l \Phi_{l+1})^* + G_l Q_{l+1} G_l^*. \tag{27}$$

In the low-rank setting, the backwards kernel is efficiently approximated given the results from above. The approximate smoothing gain is a product of a tall, a small quadratic, and a wide matrix:

$$G_l \approx \Sigma_l^{1/2} \underbrace{\Sigma_l^{*/2} \Phi_{l+1}^* \left( \Pi_{l+1}^{1/2} \right)^+}_{=: \Gamma_l \in \mathbb{R}^{r \times r}} \left( \Pi_{l+1}^{*/2} \right)^+. \tag{28}$$

During filtering, $\Gamma_l$ is saved alongside the low-rank factors of the prediction and filtering covariances. We proceed to compute a low-rank representation of the backwards-transition covariance. Consider the following singular value decomposition, truncated at the $r$-th largest singular value,

$$\left( (I - G_l \Phi_{l+1}) \Sigma_l^{1/2} \quad G_l Q_{l+1}^{1/2} \right) \approx \widehat{U}_l \widehat{D}_l \widehat{V}_l^*, \tag{29}$$

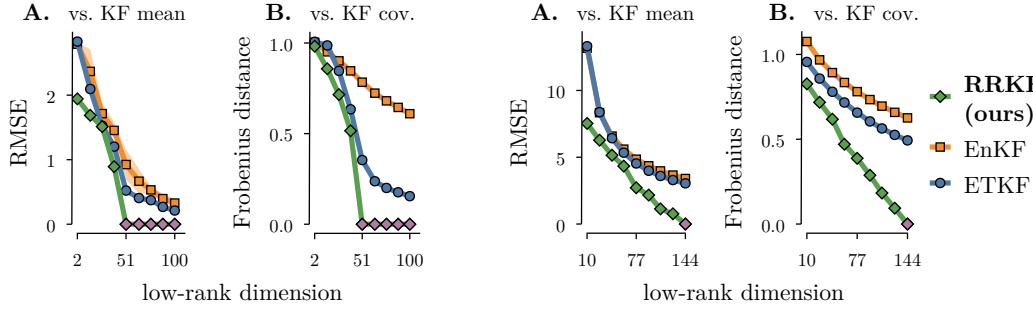

(a) *Linear advection dynamics (true rank = 51).*  (b) *London air-quality regression.*

Figure 2: *Performance of the low-rank filters on two different settings.* In the truly low-rank linear advection problem (a), the RRKF achieves the optimal estimate (◆) when $r$ exceeds the true rank of the problem. For the spatio-temporal air-quality regression (b), our method is consistently better.

where $\widehat{U}_l \in \mathbb{R}^{n \times r}$, $\widehat{D}_l \in \mathbb{R}^{r \times r}$ diagonal, $\widehat{V}_l \in \mathbb{R}^{r \times r}$, and $G_l$ as given in Eq. (28). Then a rank-$r$ factor of $P_l$ is given as

$$P_l^{1/2} = \widehat{U}_l \widehat{D}_l \in \mathbb{R}^{n \times r}. \tag{30}$$

As for filtering, in the worst case, the cost for low-rank smoothing scales quadratically with the state dimension $n$. The following proposition, which is proved in Appendix A.3, states that this can be further reduced to linear complexity in $n$, given the assumptions on the dynamics model are satisfied.

**Proposition 3.** *Given Assumption 1, computing the smoothing density $p(x_l \mid y_{1:N})$ costs $\mathcal{O}(nr^2 + r^3)$.*

Finally, it is useful to mention that realizations of the backwards process are posterior samples.

## 4   Experiments

This section evaluates the proposed method in different experimental settings. The chosen measure of quality is the distance of the approximate low-rank moments to the exact KF. We measure the mean deviations with the root-mean-squared error (RMSE) and the covariance deviations with the time-averaged relative Frobenius distances. The presented rank-reduced Kalman filter (RRKF) is compared to the ensemble Kalman filter (EnKF) and the ensemble transform Kalman filter (ETKF). All EnKF and ETKF results are given in sample statistics over 20 runs. Section 4.1 shows that for a truly low-rank system, our method recovers the KF estimate up to numerical error as soon as $r$ exceeds the true rank of the problem. Section 4.2 tests the method on a spatio-temporal GP regression problem with real data. After Section 4.3 evaluates the approximation quality for increasingly low-rank systems in a controlled experimental environment, Section 4.4 verifies the stated asymptotic cost of the method. Finally, a large-scale spatio-temporal regression problem is solved in Section 4.5. In all experiments, we compute the stationary mean and a low-rank factorization of the stationary covariance matrix of the prior and condition the stationary moments on the first measurement in the respective time-series dataset. More details on the experimental setups are given in Appendix D.

### 4.1   Linear advection model

The proposed algorithm is evaluated in a standard data-assimilation setup in which linear-advection dynamics with periodic boundary conditions are assimilated to a set of simulated data. The setup follows the description in Sakov and Oke [33] and the experimental details are additionally detailed in Appendix D. This problem has rank 51 by construction. Figure 2a shows how the exact KF estimate is recovered by our method for $r = 51$, while the ensemble methods converge according to a Monte–Carlo rate. This is particularly clear for the covariance estimate (Fig. 2a, **B.**).

### 4.2   London air-quality regression

This experiment uses hourly data from the London air-quality network [23] between January 2019 and April 2019, which amounts to measurements at 72 spatial locations and 2159 points in time. The

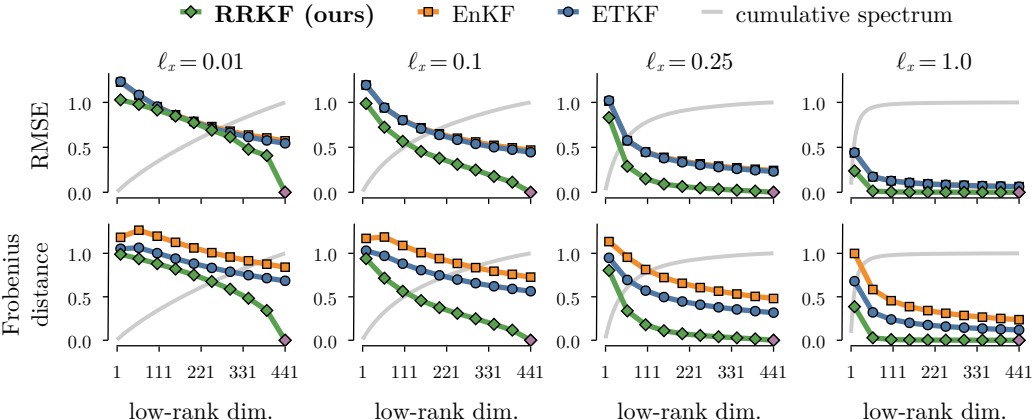

Figure 3: *How does the rate of spectral decay influence the low-rank approximation?* The larger the spatial length scale of a spatio-temporal Matérn model, the faster the spectrum decays and the faster all methods converge to the KF estimate. The first and second row compare the mean estimates and the covariance estimates, respectively. The RRKF estimate is consistently closer and recovers the KF estimate (◆) at $r = n$. The cumulative spectrum of the final-step KF covariance is shown in grey.

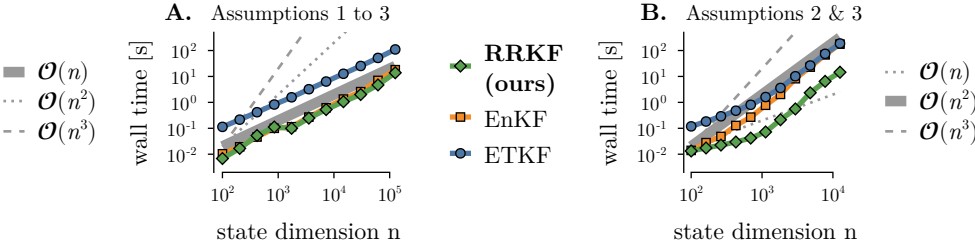

Figure 4: *Best-case and worst-case asymptotic complexities of the low-rank filters.* When all Assumptions 1 to 3 are satisfied (**A.**), the elapsed time grows linearly with the state dimension $n$. If parts of the assumptions are not met (**B.**), the cost scales quadratically with the respective dimension.

data used, together with its processing (except the train-test split), is the same as in the corresponding experiment by Hamelijnck et al. [20]. The model is a spatio-temporal Matérn-$3/2$ process with prior hyperparameters that maximize the marginal log likelihood. Figure 2b shows that—compared to the ensemble methods—our algorithm is consistently closer to the KF estimate and recovers it at $r = n$.

### 4.3 Spatio-temporal Matérn process with varying spatial lengthscale

Consider a spatio-temporal Gaussian process $\mathrm{x}(t) \sim \mathcal{GP}(0, k_t \otimes k_x)$ with covariance structure that is separable in time and space. Let both $k_t$ and $k_x$ be of the Matérn family with characteristic spatial and temporal lengthscales $\ell_t, \ell_x$ and output scales $\sigma_t^2, \sigma_x^2$. Spatio-temporal Matérn processes can be translated to the formulation in Eq. (1a), as detailed, for instance, by Solin [39]. This setting allows for varying the "low-rankness" by the choice of $\ell_x$: the larger $\ell_x$, the more information about a spatial point is prescribed by its surroundings. As $\ell_x$ approaches zero, the spatial kernel matrix gets more and more diagonal, encoding spatially independent states, and causing a barely decaying spectrum. The spectrum for large $\ell_x$ decays rapidly, making low-rank approximations more accurate for small $r$.

In the experiment, we consider a spatial domain $[0, 2] \times [0, 2] \subset \mathbb{R}^2$, which is subsampled at a uniformly-spaced grid with $\Delta_x = 0.1$ ($n = 21 \times 21 = 441$). Noisy observations of the full state trajectory, are drawn from a realization of the prior process. Figure 3 demonstrates that, for increasing spatial lengthscales, all methods converge faster to the true KF estimate. Crucially, however, our method is consistently closer to the optimal estimate and recovers it for $r = n$.

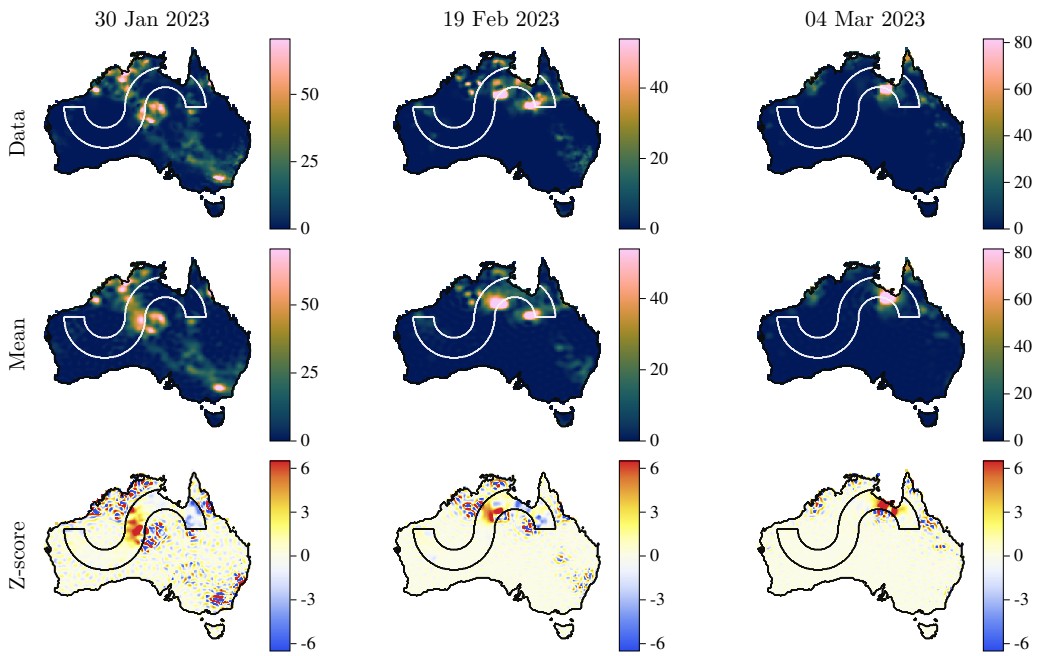

Figure 5: *Rainfall in Australia.* A spatio-temporal GP regression problem with $n = 30\,911$ state dimensions, solved with the RRKF with $r = 1000$. The columns are three different days in the time series. The rows show the data, mean estimate, and Z-scores, respectively.

## 4.4 Runtime

To demonstrate the asymptotic computational complexities given in Section 3.3, we investigate the runtimes on two different settings: The first fulfills both Assumptions 1 and 2, whereas the second only fulfills Assumption 2, which means that the cost of prediction is quadratic in the state dimension. Both problems are solved on a temporal grid of size $N = 100$ with the measurement dimension fixed at $m = 100$ and the low-rank dimension fixed at $r = 5$.

The problem from Section 4.1 serves as the best-case setting. The noise-free linear-advection dynamics amount to multiplication of the state with a circulant matrix, which is implemented efficiently using fast Fourier transform. The measurement operator amounts to array indexing. As subplot **A.** in Fig. 4 confirms, the computational cost grows linearly with the state dimension. As a worst-case example we employ a spatio-temporal Matérn process with a dense spatial kernel matrix (cf. Section 4.3). In this case, we expect the computation time to scale quadratically with the state dimension because $BB^*$ is dense. This is verified by the right plot in Fig. 4 (**B.**).

## 4.5 Large-scale spatio-temporal GP regression on rainfall data

As a final experiment, a spatio-temporal GP regression problem is solved on a large gridded analysis data set measuring rainfall in Australia. The data is provided by the Australian Water Availability Project (AWAP) [2, 24]. The number of spatial data points is $n = 30\,911$, which rules out the cubic-in-$n$ Kalman filter. The smoothing posterior is computed at $N = 40$ time points, from 25 January, 2023 through 5 March, 2023. The total number of observations is thus $n \cdot N = 1\,236\,440$. The spatio-temporal prior Matérn model is selected by maximizing the marginal log-likelihood of a lower-resolution data set. A set of 6160 spatial points is excluded from the data during filtering and smoothing in order to evaluate spatial interpolation performance. Thus, the measurement dimension is $m = n - 6160 = 24751$. Figure 5 shows the low-rank approximation to the GP posterior with $r = 1000$ low-rank dimensions. The non-observed spatial locations are framed in the horizontal-"S" shape. At measurement locations, the data is described well by the model, while a slight smoothing-out effect can be observed due to the high-frequency features of the model being truncated. At the evaluation points the interpolation aligns with the expectations regarding the simple Matérn model.

The Z-score map shows $\frac{\xi_l - y_l}{\lambda_l}$, where $\xi_l$ and $\lambda_l$ denote the smoothing mean and marginal standard deviation and $y_l$ the data point, at time $t_l$. This highlights badly calibrated uncertainty estimates, wherever the model under-/overestimates the data and is divided by a small standard deviation, in blue/red. The Z-score distribution receives further attention in Appendix E. All in all, the RRKF achieves high-quality approximate estimates while compressing the state by a factor of $\frac{30\,911}{1000} \approx 30$.

## 5 Limitations

The main limitation of the proposed approximate probabilistic inference scheme is that the truncation to $r$ dimensions cuts away covariance information, which is not accounted for. As a result, confidence *grows* as $r$ shrinks, while estimates should arguably get more uncertain instead with increased compression. In existing low-rank filters, this issue is often counteracted manually by inflating the covariance matrices, on a per-application basis [6, Section 4.4] and calls for a more principled treatment. Finding a way to keep track of that residual uncertainty information is beyond the scope of this paper. It remains unclear how to preserve the stated asymptotic complexities while doing so.

Further, all results herein are stated for linear dynamics and observation models. Extensions to non-linear models could include linearization of the according transitions, or cubature methods [34].

## 6 Conclusion

By building upon well-established knowledge about optimal compression, we have proposed an algorithm providing a principled way to balance approximation accuracy against computational cost in high-dimensional state estimation. It combines simple (truncated) singular value decompositions with recent numerical low-rank integrators of large matrix differential equations. We have offered both theoretical and empirical arguments for why it is desirable to use a deterministic algorithm when approximating large-scale probabilistic state-estimation problems.

## Acknowledgments and Disclosure of Funding

The authors gratefully acknowledge financial support by the German Federal Ministry of Education and Research (BMBF) through Project ADIMEM (FKZ 01IS18052B), and financial support by the European Research Council through ERC StG Action 757275 / PANAMA; the DFG Cluster of Excellence "Machine Learning - New Perspectives for Science", EXC 2064/1, project number 390727645; the German Federal Ministry of Education and Research (BMBF) through the Tübingen AI Center (FKZ: 01IS18039A); and funds from the Ministry of Science, Research and Arts of the State of Baden-Württemberg. Jörg Nick is funded by the Deutsche Forschungsgemeinschaft (DFG, German Research Foundation) – Project-ID 258734477 – SFB 1173. Filip Tronarp was partially supported by the Wallenberg AI, Autonomous Systems and Software Program (WASP) funded by the Knut and Alice Wallenberg Foundation. The authors thank the International Max Planck Research School for Intelligent Systems (IMPRS-IS) for supporting Jonathan Schmidt. The authors also thank Nathanael Bosch and Christian Lubich for many valuable discussions and for helpful feedback.

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
