# The Rank-Reduced Kalman Filter: Approximate Dynamical-Low-Rank Filtering In High Dimensions
## Appendix

## A Proofs

### A.1 Proof of Proposition 1

*Proof.* By standard results on Gaussian conditioning and marginalization

$$z \mid y \sim \mathcal{N}(K_z(y - C\mu), \Sigma_z), \tag{A.1a}$$

$$y \sim \mathcal{N}(C\mu, S), \tag{A.1b}$$

where

$$\Sigma_z^{-1} = I + (R^{-1/2}C\Pi^{1/2})^*R^{-1/2}C\Pi^{1/2}, \tag{A.2a}$$

$$K_z = \Sigma_z(R^{-1/2}C\Pi^{1/2})^*R^{-1/2}, \tag{A.2b}$$

$$R^{-1/2}SR^{-*/2} = R^{-1/2}C\Pi^{1/2}(R^{-1/2}C\Pi^{1/2})^* + I. \tag{A.2c}$$

Substituting for the singular value decomposition of $(R^{-1/2}C\Pi^{1/2})^*$ gives the conditional covariance

$$\Sigma_z = \left(I + UD^2U^*\right)^{-1} = \left(U(I + D^2)U^*\right)^{-1} = U(I + D^2)^{-1}U^*, \tag{A.3}$$

the Kalman gain

$$K_z = U(I + D^2)^{-1}U^*UDV^*R^{-1/2} = U(I + D^2)^{-1}DV^*R^{-1/2}, \tag{A.4}$$

and the marginal measurement covariance matrix

$$R^{-1/2}SR^{-*/2} = VD^2V^* + I. \tag{A.5}$$

This gives the result on marginalization and conditioning. To obtain the expression for the marginal likelihood, which is given by

$$\mathcal{N}(y; C\mu, S) = -\frac{m}{2}\log 2\pi - \frac{1}{2}\log|S| - \frac{1}{2}(y - C\mu)^*S^{-1}(y - C\mu). \tag{A.6}$$

The log determinant is given by

$$\log|S| = \log\left|R^{1/2}(VD^2V^* + I)R^{*/2}\right| = 2\log\left|R^{1/2}\right| + \log\left|VD^2V^* + I\right|. \tag{A.7}$$

Let $V_k$ be the $k$-th column vector of $V$ then it is an eigenvector of $VD^2V^* + I$ with eigenvalue $D_{kk}^2 + 1$. Furthermore, any vector in the orthogonal complement to the column space of $V$ is also an eigenvector with eigenvalue 1. Therefore,

$$\log\left|VD^2V^* + I\right| = \sum_{k=1}^{r}\log(D_{kk}^2 + 1). \tag{A.8}$$

It remains to obtain the desired expression for the quadratic form. Start by inserting the expression for S

$$\begin{aligned}(y - C\mu)^*S^{-1}(y - C\mu) &= (y - C\mu)^*R^{-*/2}(VD^2V^* + I)^{-1}R^{-1/2}(y - C\mu) \\ &= e^*(VD^2V^* + I)^{-1}e.\end{aligned} \tag{A.9}$$

Now $e^*(VD^2V^* + I)^{-1}e = e^*b$, where we define the vector $b$ such that

$$(VD^2V^* + I)b = e. \tag{A.10}$$

Multiplying from the left by $V^*$ gives

$$(D^2 + I)V^*b = V^*e, \tag{A.11}$$

therefore,

$$V^*b = (D^2 + I)^{-1}V^*e. \tag{A.12}$$

Inserting this in the original definition of $b$ gives

$$VD^2(D^2 + I)^{-1}V^*e + b = e, \tag{A.13}$$

from which it follows that

$$e^*(VD^2V^* + I)^{-1}e = e^*b = \|e\|^2 - e^*VD(D^2 + I)^{-1}DV^*e. \tag{A.14}$$

$\square$

## A.2 Proof of Proposition 2

In the following, the computational complexity of the low-rank filtering recursion is analyzed in detail, which proves Proposition 2 and Corollary 2. In the best case, the cost of the proposed filtering algorithms scales linearly in the state dimension $n$ and the measurement dimension $m$. For this to hold, we assume that

    (a) the maps $x \mapsto Ax$, $x \mapsto \Phi x$, and $x \mapsto BB^*x$ can be evaluated in $\mathcal{O}(n)$,

    (b) the map $x \mapsto Cx$ can be evaluated in $\mathcal{O}(m)$, and

    (c) the map $x \mapsto R^{-1/2}x$ and the log-determinant $\log|R^{1/2}|$ can be evaluated in $\mathcal{O}(m)$.

We refer to the situation in which (a) or (b) do not apply as the "worst case". Assumption (c) is taken for granted as the measurement-noise covariance $R$ is often a diagonal matrix, which implies that sensor errors are uncorrelated. This is not only realistic but also commonly imposed in modelling.

*Proof.* The best-case cost of the approximate filtering scales linear in the state dimension $n$ and the measurement dimension $m$. In the worst case, the complexity scales quadratically in $n$ and $m$.

**Prediction** We begin by analyzing the cost for the approximate integration of the low-rank process-noise covariance $Q^{1/2}$. This amounts to the cost of the dynamical low-rank approximation (DLRA) algorithm for a symmetric Lyapunov equation. Appendix C gives a detailed description of how this algorithm is used as part of the RRKF recursions. Let the matrix-valued flow field be $F(Q) = AQ + QA^* + BB^*$. Given an initial factorization $Q_0 \approx Y_0 = U_0 D_0^2 U_0^*$, with $U_0 \in \mathbb{R}^{n \times r}$ and $D_0 \in \mathbb{R}^{r \times r}$, the cost for integrating this matrix equation using DLRA is as follows:

    1. **K-step**

        (a) Flow-field evaluation

$$F(t, K(t)U_0^*)U_0 = A(K(t)U_0^*)U_0 + K(t)U_0^*A^*U_0 + BB^*U_0 \tag{A.15a}$$
$$= AK(t) + K(t)(U_0^*(A^*U_0)) + BB^*U_0, \tag{A.15b}$$

        (b) QR factorization of tall $n \times r$ matrix

$$\mathrm{QR}\left(K(t+h)\right), \tag{A.16}$$

        (c) Compute

$$M = U_h^*U_0. \tag{A.17}$$

    2. **S-step**

        (a) Compute initial $D(t_0)$

$$D(t_0) = MD_0M^*, \tag{A.18}$$

        (b) Flow-field evaluation

$$U_h^*F(t, U_hD(t)U_h^*)U_h = U_h^*\left(AU_hD(t)U_h^* + U_hD(t)U_h^*A^* + BB^*\right)U_h \tag{A.19a}$$
$$= U_h^*(A(U_hD(t))) + (D(t)U_h^*)(A^*U_h) + U_h^*(BB^*U_h). \tag{A.19b}$$

Table A.1: Time complexity of reduced-rank filtering: The prediction step

| Eq. | Operation | Best case | Worst case |
|------|-----------|-----------|------------|
| (A.15) | $AK(t)$ 
 $K(t)(U_0^*(A^*U_0))$ 
 $BB^*U_0$ | $\mathcal{O}(nr)$ 
 $\mathcal{O}(nr^2)$ 
 $\mathcal{O}(nr)$ | $\mathcal{O}(n^2r)$ 
 $\mathcal{O}(n^2r + nr^2)$ 
 $\mathcal{O}(n^2r)$ |
| (A.16) | $\mathrm{QR}\left(K(t+h)\right)$ | $\mathcal{O}(nr^2)$ | $\mathcal{O}(nr^2)$ |
| (A.17) | $M = U_h^*U_0$ | $\mathcal{O}(nr^2)$ | $\mathcal{O}(nr^2)$ |
| (A.18) | $D(t_0) = MD_0M^*$ | $\mathcal{O}(r^3)$ | $\mathcal{O}(r^3)$ |
| (A.19) | $U_h^*(A(U_hD(t)))$ 
 $(D(t)U_h^*)(A^*U_h)$ 
 $U_h^*(BB^*U_h)$ | $\mathcal{O}(nr^2)$ 
 $\mathcal{O}(nr^2)$ 
 $\mathcal{O}(nr^2)$ | $\mathcal{O}(n^2r + nr^2)$ 
 $\mathcal{O}(n^2r + nr^2)$ 
 $\mathcal{O}(n^2r + nr^2)$ |
| (A.20) | $D_l = D_l^{1/2}D_l^{*/2}$ | $\mathcal{O}(r^3)$ | $\mathcal{O}(r^3)$ |
| (A.21) | $\Phi_l\mu_{l-1}$ | $\mathcal{O}(n)$ | $\mathcal{O}(n^2)$ |
| (A.22) | $\Phi_l\Sigma_{l-1}^{1/2}$ 
 $(\Phi_l\Sigma_{l-1}^{1/2} \quad Q_l^{1/2}) \approx \tilde{U}_l\tilde{D}_l\tilde{V}_l^*$ | $\mathcal{O}(nr)$ 
 $\mathcal{O}(nr^2)$ | $\mathcal{O}(n^2r)$ 
 $\mathcal{O}(nr^2)$ |

The above steps 1. and 2. are repeated according to how many DLRA integration steps are performed in the respective prediction step. Therefore, their cost has to be multiplied by that (typically small) constant.

3. Using DLRA integration we obtain a low-rank factorization of the process-noise covariance matrix $Q_l \approx Y_l = U_lD_lU_l^2$. It remains to compute a matrix square root $Q_l^{1/2} = U_lD_l^{1/2}$ and thus a matrix square root of $D_l \in \mathbb{R}^{r \times r}$

$$D_l = D_l^{1/2}D_l^{*/2}. \tag{A.20}$$

Now, having obtained $Q_l^{1/2}$, we proceed to

4. predict the mean
$$\mu_l^- = \Phi_l\mu_{l-1}, \tag{A.21}$$

5. and the low-rank factor $\Pi_l^{1/2}$ of the predicted covariance matrix. This amounts to building the rank-$2r$ square-root factor and truncating it at its $r$-th largest singular value (i.e., a truncated SVD of a tall $n \times 2r$ matrix):
$$\left(\Phi_l\Sigma_{l-1}^{1/2} \quad Q_l^{1/2}\right) \approx \tilde{U}_l\tilde{D}_l\tilde{V}_l^*. \tag{A.22}$$

The computational complexities of the respective steps can be found in Table A.1.

**Correction step** Now, the cost of the correction step is analyzed. First, we compute an SVD of a wide $r \times m$ matrix
$$(R^{-1/2}C\Pi_l^{1/2})^* = U_lD_lV_l^*. \tag{A.23}$$

Given this decomposition and the whitened residual
$$e_l = R^{-1/2}(y_l - C\mu_l^-), \tag{A.24}$$

we proceed to

1. update the mean
$$\Delta\mu_l = K_le_l = \Pi_l^{1/2}(U_l((I + D_l^2)^{-1}(D_l(V_l^*e_l)))), \tag{A.25}$$

Table A.2: Time complexity of reduced-rank filtering: The correction step

| Eq. | Operation | Best case | Worst case |
|---|---|---|---|
| (A.23) | $R^{-1/2}(C\Pi_l^{1/2})$ | $\mathcal{O}(mr)$ | $\mathcal{O}(m^2 r)$ |
| | $(R^{-1/2}C\Pi_l^{1/2})^* = U_l D_l V_l^*$ | $\mathcal{O}(mr^2)$ | $\mathcal{O}(mr^2)$ |
| (A.24) | $e_l = R^{-1/2}(y_l - C\mu_l^-)$ | $\mathcal{O}(m)$ | $\mathcal{O}(mn + m^2)$ |
| (A.25) | $\Pi_l^{1/2}(U_l((I + D_l^2)^{-1}(D_l(V_l^* e_l))))$ | $\mathcal{O}(mr + nr^2)$ | $\mathcal{O}(mr + nr^2)$ |
| (A.26) | $\Pi_l^{1/2}(U_l(I + D_l^2)^{-1/2})$ | $\mathcal{O}(nr^2)$ | $\mathcal{O}(nr^2)$ |
| (23) | $\log|R^{1/2}|$ | $\mathcal{O}(m)$ | $\mathcal{O}(m)$ |
| | $\sum_{k=1}^r \log((D_l)_{kk}^2 + 1)$ | $\mathcal{O}(r)$ | $\mathcal{O}(r)$ |
| | $\|e_l\|^2$ | $\mathcal{O}(r)$ | $\mathcal{O}(r)$ |
| | $(e_l^* V_l)(D_l((D_l^2 + I)^{-1} D_l))(V_l^* e_l)$ | $\mathcal{O}(mr)$ | $\mathcal{O}(mr)$ |

2. and compute the low-rank factor of the filtering covariance

$$\Sigma_l^{1/2} = \Pi_l^{1/2}(U_l(I + D_l^2)^{-1/2}). \tag{A.26}$$

Finally, the marginal predictive log-likelihood $\log p(y_l \mid y_{1:l-1})$ is computed as in Eq. (23).

The computational complexities of the respective steps can be found in Table A.2. Together, Tables A.1 and A.2 prove the stated asymptotic complexities of the low-rank filtering recursion. $\square$

## A.3 Proof of Proposition 3

This section analyzes the computational complexity of the low-rank smoothing recursion in detail, which proves Proposition 3.

*Proof.* The best-case cost of the approximate smoothing scales linear in the state dimension $n$. In the worst case, the complexity is quadratic in $n$.

**Approximate backwards kernel**  Factorizations of $\Sigma_{l-1}^{1/2}$ and $\Pi_l^{1/2}$ are already given from filtering. For the smoothing gain, it remains to compute

$$\Gamma_l = \Sigma_l^{*/2}(\Phi_{l+1}^*(\Pi_{l+1}^{1/2})^+). \tag{A.27}$$

Then, we proceed to compute the shift vector

$$v_l = \mu_l - G_l \mu_{l+1}^- = \mu_l - \Sigma_l^{1/2}(\Gamma_l((\Pi_{l+1}^{*/2})^+ \mu_{l+1}^-)), \tag{A.28}$$

and the low-rank factor $P_l^{1/2}$ of the backwards-process-noise covariance matrix. Therefore, a truncated SVD of a tall $n \times 2r$ matrix is computed:

$$\left( (I - G_l \Phi_{l+1})\Sigma_l^{1/2} \quad G_l Q_{l+1}^{1/2} \right) \approx \widehat{U}_l \widehat{D}_l \widehat{V}_l^*. \tag{A.29}$$

**Backwards prediction (smoothing)**  Smoothing amounts to consecutive predictions with the backwards kernel. For this prediction, the low-rank factor of the process-noise covariance matrix does not come from DLRA, but can be directly computed from the smoothing gain and the process-noise covariance matrix (see Eqs. (29) and (30)).

The smoothing mean is therefore given as

$$\xi_l = G_l \xi_{l+1} + v_l = \Sigma_l^{1/2}(\Gamma_l((\Pi_{l+1}^{*/2})^+ \xi_{l+1})) + v_l, \tag{A.30}$$

Table A.3: Time complexity of reduced-rank smoothing

| Eq. | Operation | Best case | Worst case |
|---|---|---|---|
| (A.27) | $\Gamma_l = \Sigma_{l-1}^{*/2}(\Phi_l^*(\Pi_l^{1/2})^+)$ | $\mathcal{O}(nr^2)$ | $\mathcal{O}(n^2r + nr^2)$ |
| (A.28) | $G_l\mu_{l+1}^- = \Sigma_l^{1/2}(\Gamma_l((\Pi_{l+1}^{*/2})^+\mu_{l+1}^-))$ | $\mathcal{O}(nr + r^2)$ | $\mathcal{O}(nr + r^2)$ |
| (A.29) | $G_l\Phi_{l+1}\Sigma_l^{1/2} = \Sigma_l^{1/2}(\Gamma_l((\Pi_{l+1}^{*/2})^+(\Phi_{l+1}\Sigma_l^{1/2})))$ | $\mathcal{O}(nr^2 + r^3)$ | $\mathcal{O}(n^2r + nr^2 + r^3)$ |
| | $G_lQ_l^{1/2} = \Sigma_{l-1}^{1/2}(\Gamma_l((\Pi_l^{*/2})^+Q_l^{1/2}))$ | $\mathcal{O}(nr^2 + r^3)$ | $\mathcal{O}(nr^2 + r^3)$ |
| | $((I - G_l\Phi_{l+1})\Sigma_l^{1/2} \quad G_lQ_{l+1}^{1/2}) \approx \widehat{U}_l\widehat{D}_l\widehat{V}_l^*$ | $\mathcal{O}(nr^2)$ | $\mathcal{O}(nr^2)$ |
| (A.30) | $G_l\xi_l = \Sigma_{l-1}^{1/2}(\Gamma_l((\Pi_l^{*/2})^+\xi_l))$ | $\mathcal{O}(nr + r^2)$ | $\mathcal{O}(nr + r^2)$ |
| (A.31) | $G_l\Lambda_{l+1}^{1/2} = \Sigma_l^{1/2}(\Gamma_l((\Pi_{l+1}^{*/2})^+\Lambda_{l+1}^{1/2}))$ | $\mathcal{O}(nr^2 + r^3)$ | $\mathcal{O}(nr^2 + r^3)$ |
| | $(G_l\Lambda_{l+1}^{1/2} \quad P_l^{1/2}) \approx \bar{U}_l\bar{D}_l\bar{V}_l^*$ | $\mathcal{O}(nr^2)$ | $\mathcal{O}(nr^2)$ |

and the low-rank factor of the smoothing covariance as $\Lambda_l^{1/2} = \bar{U}_l\bar{D}_l$, where

$$\left(G_l\Lambda_{l+1}^{1/2} \quad P_l^{1/2}\right) \approx \bar{U}_l\bar{D}_l\bar{V}_l^* \tag{A.31}$$

is the truncated SVD of a tall $n \times 2r$ matrix.

The computational complexities of the respective steps can be found in Table A.3, which shows that—in the best case—the cost of approximate low-rank smoothing never exceeds $\mathcal{O}(nr^2 + r^3)$, as stated by Proposition 3. It also shows that in the worst case, the complexity is quadratic in $n$. $\quad\square$

## B  Efficient inference for the case r > m

This section gives an inference scheme that can be used in the correction step in case the low-rank dimension $r$ exceeds the measurement dimension $m$ at a time point $t_l$. This is not generally the designated use case of the algorithm, since in most applications, we assume $r \ll m \leq n$. However, if at a given time point $t_l$, there are less than usual measurements available (e.g. due to missing data at that time), it is still useful to have an inference scheme in place for this case. This is provided by the following proposition.

**Proposition B.1.** *Let $m < r \leq n$ and $\Pi_l^{1/2} \in \mathbb{R}^{n \times r}$ and $R^{1/2} \in \mathbb{R}^{m \times m}$. An approximate update is then obtained according to the standard Kalman filter update rules:*

$$S_l = C\Pi_l^{1/2}(C\Pi_l^{1/2})^* + R^{1/2}R^{*/2}, \tag{B.1a}$$

$$K_l = \Pi_l^{1/2}(C\Pi_l^{1/2})^*S^+, \tag{B.1b}$$

$$\Delta\mu_l = K_l(y_l - C\mu_l), \tag{B.1c}$$

$$\Sigma_l = \Pi_l - K_lS_lK_l^*. \tag{B.1d}$$

*The marginal measurement covariance, $S_l$, is obtained by a singular value decomposition*

$$U_l^sD_l^s(V_l^s)^* = \left(C\Pi_l^{1/2} \quad R^{1/2}\right), \tag{B.2a}$$

$$S = U_l^s(D_l^s)^2(U_l^s)^*, \tag{B.2b}$$

*where the matrix on the right-hand side of the first equation is $m \times (r + m)$. Hence, $U_l^s \in \mathbb{R}^{m \times (r+m)}$, and $D_l^s, V_l^s \in \mathbb{R}^{(r+m) \times (r+m)}$. The Moore–Penrose pseudoinverse $S_l^+$ is then given by*

$$S_l^+ = U_l^s(D_l^s)^{-2}(U_l^s)^*. \tag{B.3}$$

*The gain matrix is thus obtained as*

$$\widetilde{K}_l = (C\Pi_l^{1/2})^*U^s(D^s)^{-1}, \tag{B.4a}$$

$$K_l = \Pi_l^{1/2}\widetilde{K}_l(D^s)^{-1}(U^s)^*, \tag{B.4b}$$

*where $\widetilde{K}_l \in \mathbb{R}^{r \times (r+m)}$. The updated covariance is obtained by*

$$
\begin{aligned}
\Sigma_l &= \Pi_l - K_l S_l K_l^* \\
&= \Pi_l - \Pi_l^{1/2} \widetilde{K}_l (D_l^s)^{-1} (U_l^s)^* U_l^s D_l^s (U_l^s)^* (\Pi_l^{1/2} \widetilde{K}_l (D_l^s)^{-1} (U_l^s)^*)^* \\
&= \Pi_l - \Pi_l^{1/2} \widetilde{K}_l (\Pi_l^{1/2} \widetilde{K}_l)^* \\
&= \Pi_l^{1/2} \big(I - \widetilde{K}_l \widetilde{K}_l^*\big) \Pi_l^{*/2}.
\end{aligned}
\tag{B.5}
$$

*Consider the singular value decomposition of $\widetilde{K}_l$*

$$
\widetilde{K}_l = U_l^k D_l^k (V_l^k)^*,
\tag{B.6}
$$

*where $U_l^k, D_l^k \in \mathbb{R}^{r \times r}$ and $V_l^k \in \mathbb{R}^{r \times r+m}$. The covariance update in low-rank form is then given by*

$$
\begin{aligned}
\Sigma^{1/2} &= \Pi^{1/2} \big(I - U_l^k (D_l^k)^2 (U_l^k)^*\big)^{1/2}, \\
&= \Pi^{1/2} \big(U_l^k (U_l^k)^* - U_l^k (D_l^k)^2 (U_l^k)^*\big)^{1/2}, \\
&= \Pi^{1/2} U_l^k \big(I - (D_l^k)^2\big)^{1/2}.
\end{aligned}
\tag{B.7}
$$

*The marginal predictive log likelihood $p(y_l \mid y_{1:l-1}) = \mathcal{N}(C\mu_l^-, S_l)$ can be cheaply evaluated given the factorization of $S_l$ from Eq. (B.2).*

## C Dynamical-low-rank approximation algorithm for Lyapunov equations

Following Ceruti and Lubich [8], we give the procedure for one DLRA integration step, adapted to our specific case of a symmetric Lyapunov equation

$$
F(t, Q(t)) = AQ(t) + Q(t)A^* + B(t)B(t)^*.
\tag{C.1}
$$

Let an initial rank-$r$ factorization $Q_0 \approx Y_0 = U_0 D_0 U_0^*$, with an orthogonal matrix $U_0 \in \mathbb{R}^{n \times r}$ and a symmetric matrix $D_0 \in \mathbb{R}^{r \times r}$, be given at time $t_0$. For a temporal step size $h$, a rank-$r$ factorization $Q(t+h) \approx Y_h = U_h D_h U_h^*$ at the next integration step $t_0 + h$ is computed as follows.

**K-step:** Update $U_0 \to U_h$.
Integrate from $t = t_0$ to $t_0 + h$ the $n \times r$ matrix differential equation

$$
\dot{K}(t) = F(t, K(t)U_0^*)U_0, \qquad K(t_0) = U_0 D_0.
\tag{C.2}
$$

Orthogonalize $K(t+h)$ by computing a QR decomposition, yielding the orthogonal matrix $U_h$. Then, compute the $r \times r$ matrix $M = U_h^* U_0$.

**S-step:** Update $D_0 \to D_h$.
Integrate from $t = t_0$ to $t_0 + h$ the $r \times r$ matrix differential equation

$$
\dot{D}(t) = U_h^* F(t, U_h D(t) U_h^*) U_h, \qquad D(t_0) = M D_0 M^*,
\tag{C.3}
$$

and set $D_h = D(t+h)$

**Remark C.1.** *The terms "K-step" and "S-step" can be confusing due to conflicting notation conventions in the Kalman-filtering and dynamical-low-rank-approximation literature, respectively. The matrix $K$ and the term "S-step" are not related to the Kalman gain $K$ and the marginal measurement covariance matrix $S$ from the Kalman filter update step described in Section 3.2 and Appendix B.*

The $n \times r$ and $r \times r$ matrix differential equations in the K-step and S-step can be solved using a standard numerical integrator, e.g., a Runge-Kutta method or an exponential integrator.

It is useful to note that the matrix equation in the S-step itself is a symmetric Lyapunov equation:

$$
\begin{aligned}
\dot{D}(t) &= U_h^* F(t, U_h D(t) U_h^*) U_h && \text{(C.4a)} \\
&= U_h^* \big(A U_h D(t) U_h^* + U_h D(t) U_h^* A^* + BB^*\big) U_h && \text{(C.4b)} \\
&= (U_h^* A U_h) D(t) + D(t) (U_h^* A U_h)^* + U_h^* BB^* U_h && \text{(C.4c)} \\
&= A_D D(t) + D(t) A_D^* + B_D B_D^*, && \text{(C.4d)}
\end{aligned}
$$

where the final step defines the parameters of the Lyapunov equation for $D(t)$ as $A_D = U_h^* A U_h$ and $B_D = U_h^* B$. Since $D(t) \in \mathbb{R}^{r \times r}$ is small, this equation can be solved exactly with little computational cost using matrix-fraction decomposition [40, 3].

# D   Details on experimental setups

This section gives more details regarding the experimental setups used in Section 4. For all results of the proposed RRKF algorithm, only a single DLRA step was used in the prediction step to compute the low-rank factorization of the process noise covariance matrix $Q^{1/2}$.

## D.1   Linear advection model

This section provides more details on the data assimilation setting that is considered in Section 4.1.

We consider a spatial grid of $n = 1024$ uniformly spaced points and a temporal grid of 800 uniformly spaced time points. Unit step sizes $\Delta t = \Delta x = 1$ are assumed on the spatio-temporal grid. To generate an initial ground-truth state we sample an initial sinusoidal curve $\psi(0)$ according to the description by Sakov and Oke [33]

$$\psi_i(0) = \sum_{k=0}^{25} a_k \sin\left(\frac{2\pi k}{1000} i + \varphi_k\right), \tag{D.1}$$

where $i = 1, \dots, 1024$ is an index into the spatial grid and $a_k \sim \mathrm{Unif}(0, 1)$, $\varphi_k \sim \mathrm{Unif}(0, 2\pi)$. The initial state is normalized as described by Sakov and Oke [33]. To generate a ground-truth trajectory from this initial state, the linear-advection dynamics $\frac{\partial \psi}{\partial t} = -\alpha \frac{\partial \psi}{\partial x}$ are simulated on the finite spatial grid. We assume constant unit velocity $\alpha = 1$ and periodic boundary conditions in space. As data, 10 equidistant state components are observed every 5 time steps. Each observation is corrupted by additive Gaussian noise.

An initial ensemble of size $r = n$ is built by successive sampling according to Eq. (D.1). The exact sampling process follows the more detailed description by Evensen [14, Section 3.1]. From this $\mathbb{R}^{n \times n}$ ensemble matrix, the sample covariance matrix is computed and used as the initial covariance for the Kalman filter. The initial factorization of the RRKF is obtained by a spectral decomposition of the sample covariance matrix truncated at the $r$-th largest eigenvalue. Selecting the $r$ first ensembles serves as an initial ensemble for the EnKF and the ETKF.

Figure 2a shows the deviations of the individual low-rank approximations to the optimal KF estimate on the described data assimilation problem.

## D.2   London air-quality regression

The experimental setup and the data used in Section 4.2 is provided by Hamelijnck et al. [20]. The model is selected via the log-likelihood estimation of the *exact* Kalman filter. The RMSE of the Kalman filter mean to the test data is 9.96791 (cf. Hamelijnck et al. [20]). For $r = n$ the RRKF also obtains this RMSE up to numerical error, as shown in Fig. 2b.

## D.3   Spatio-temporal Matérn process with varying spatial lengthscale

Section 4.3 evaluates the low-rank approximation quality in approximate on-model spatio-temporal GP regression with varying strength of interaction between the state components. In this experiment, we consider a spatio-temporal GP with separable covariance structure, as described in Section 4.3 with hand-picked hyperparameters. The time domain is a uniform grid on the interval $[0.1, 10]$ with step size $\Delta t = 0.1$. The spatial domain is a uniform grid on $[0, 2] \times [0, 2]$ with step size $\Delta x = 0.1$.

Over the experiment, the spatial lengthscale is varied in order to evaluate low-rank approximation quality given how much correlation between the respective state components is encoded in the prior. The smaller the spatial lengthscale, the less interactions between the spatial point and the worse the low-rank approximation is expected to be for small $r$. We test the values $\ell_x \in \{0.01, 0.1, 0.25, 1.0\}$. For each of those values, we proceed as follows. First, a realization of the prior dynamics is drawn from the spatio-temporal Matérn process. At every temporal grid point, the entire state vector of the prior draw at that point is corrupted by additive Gaussian noise to generate data. Then, the Kalman filter estimate and the respective low-rank filtering estimates are computed for increasing values of $r$. Figure 3 shows for increasing spatial lengthscales and varying $r$ (i) the resulting RMSE of the approximate filter means to the Kalman filter mean and (ii) the time-averaged Frobenius distance between the approximate filter covariance matrices to the Kalman filter covariance matrix.

## D.4 Runtime

This section details the respective experimental setups used to evaluate the best-case and worst-case computational complexity of the approximate low-rank filters with respect to the state dimension $n$, as described in Section 4.4. We analyze the asymptotic computational complexity with respect to the state dimension $n$ both for the linear-in-$n$ case (Fig. 4, left) and for the quadratic-in-$n$ case (Fig. 4, right). The elapsed wall time is always measured using the BenchmarkTools.jl software package[3] with default settings, which computes sample statistics over multiple runs automatically to prevent distortions by background processes. The setups for both analyzed cases differ only marginally from already encountered setups.

**Linear-in-$n$ case**  We use a setting that is similar to what is described in Section 4.1 and Appendix D.1. The state dimension is varied in order to evaluate computational complexity with respect to $n$. The measurement dimension is fixed at $m = 100$ and the low-rank dimension at $r = 5$. The temporal domain is a grid on the interval $[0, 100]$ with unit step size $\Delta t = 1$. Noisy observations are generated from the ground truth trajectory at every 5 time points.

The discretized linear advection dynamics with periodic boundary conditions amount to multiplication with a circulant matrix. The shift direction depends on the advection velocity, which we chose as $\alpha = 1$. Multiplication with a circulant matrix can be carried out in linear complexity via multiplication in the Fourier space. Since there is no process noise assumed, the matrix $BB^*$ is the zero matrix. Hence, Assumption 1 is satisfied by the dynamics model.

The measurement operator selects $m = 100$ uniformly spaced spatial points from the state vector and the measurement noise covariance matrix is a diagonal matrix. Hence, the measurement model fulfills Assumptions 2 and 3.

**Quadratic-in-$n$ case**  Here, we use a setting that is similar to what is described in Section 4.3 and Appendix D.3. The measurement dimension is fixed at $m = 100$ and the low-rank dimension at $r = 5$. The temporal domain is a grid on the interval $[0, 10]$ with step size $\Delta t = 0.1$. The spatial domain are $n$ equidistant points in an interval in $\mathbb{R}$, where $n$ varies. The generated data are noisy observations from a ground-truth realization of the prior at 20 time points, each measuring $m = 100$ state components. We evaluate the runtime of solving a spatio-temporal regression problem with this prior and the generated data.

Assumption 1 is *not* fulfilled by the dynamics model in this setting, in that evaluating the map $x \mapsto BB^*x$ costs $\mathcal{O}(n^2)$. By modeling spatial diffusion with a dense spatial kernel matrix, the inhomogeneity in the Lyapunov equation for the process-noise covariance matrix is also a dense matrix. Concretely, $BB^* = (\widetilde{B}\mathcal{K}_x^{1/2})(\widetilde{B}\mathcal{K}_x^{1/2})^*$, where $\widetilde{B}$ is the temporal dispersion matrix and $\mathcal{K}_x$ denotes the spatial kernel matrix. Since $\mathcal{K}_x$ is dense, so is $BB^*$, which violates a part of Assumption 1. The cost of the low-rank prediction step thus scales quadratically in the state dimension $n$.

The observation model is a sparse selection operator that projects the state onto $m = 100$ points and the measurement noise covariance is a diagonal matrix. Hence, Assumptions 2 and 3 are still fulfilled by the observation model.

Figure 4 demonstrates that the theoretical complexities are empirically verified.

## D.5 Large-scale spatio-temporal GP regression on rainfall data

This section provides further details on the large-scale low-rank approximation to spatio-temporal GP regression, described in Section 4.5.

We select the prior model via the log-likelihood estimate of the RRKF, as given in Eq. (23). For the model selection, the data set is distinct from the data used in the filtering/smoothing problem (Fig. 5). We use the time period from 25 January, 2010 through 5 March, 2010 and each data point is downsampled to a lower spatial resolution using cubic-spline interpolation. This reduces the state dimension to $n = 4350$ spatial points during model selection. Further, for model selection the low-rank dimension is set to $r = 200$.

---

[3]`https://github.com/JuliaCI/BenchmarkTools.jl`

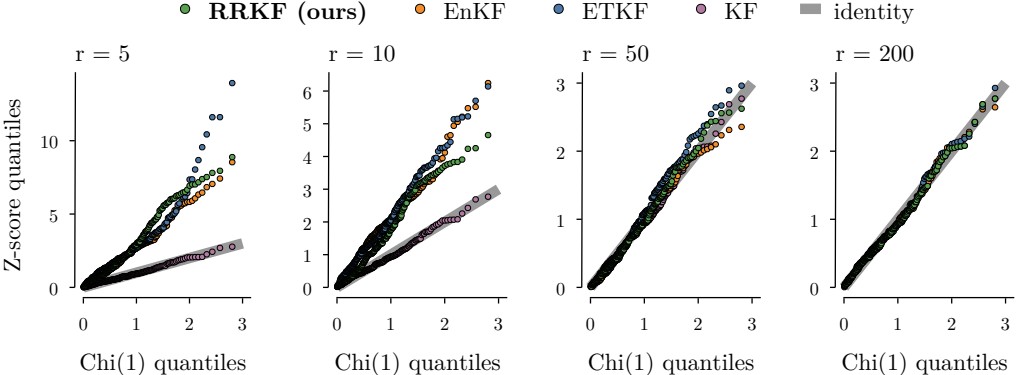

Figure E.1: *Z-score distribution for the low-rank filters with varying low-rank dimensions $r$.* In this on-model GP regression problem, we assume the Z-scores to be Chi(1) distributed. For the approximate low-rank filters, the Z-score distribution has too much mass in the high regimes, indicating overconfident estimates. For $r = n$, the Z-scores of the RRKF and the KF align.

# E  Z-scores of (approximate) Gaussian state estimates

This sections investigates a limitation of the proposed algorithm, which is addressed in Section 5. When truncating covariance information between the state components, we expect this to be reflected by a *higher* uncertainty in the resulting estimate. However, the algorithm does not account for that missing information and tends to return overconfident estimates for small values of $r$.

We analyze the distribution of the Z-scores of Gaussian state estimates on a Gaussian model. We expect the absolute values of the Z-scores to be distributed according to a Chi(1) distribution. A spatio-temporal GP regression problem in an on-model setting, similar to the setting described in Appendix D.3, is solved to investigate this. A ground truth realization is drawn from a spatio-temporal Matérn-½ process in the temporal domain $[0, 50]$ with a step size of $\Delta t = 0.1$. Data is generated by adding zero-mean Gaussian noise to $m = 150$ state components at 100 randomly sampled time points. The spatial grid is a uniformly spaced grid in the interval $[0, 20]$ with spatial step size of $\Delta x = 0.1$.

Figure E.1 visualizes the distribution of the vector of Z-scores $\frac{\mu_N - \mathrm{x}_N^\star}{\sigma_N}$, where $\mu_N$ and $\sigma_N$ are the final-step filtering/smoothing mean and standard deviation, respectively. $\mathrm{x}_N^\star$ is the ground-truth state at time $t_N$. The Z-scores are computed for increasing values of $r$. The first $r = 50$ eigenvalues of the final-step Kalman filter covariance matrix account for around $97.8\%$ of the spectrum. No covariance inflation [6, Section 4.4] is used for the ensemble methods.

It becomes apparent that—especially for very small values of $r$—there are significantly more high Z-scores than expected. This indicates that too many states are estimated poorly and divided by a small standard deviation. In their naïve implementation, all examined low-rank algorithms exhibit this behavior. For the RRKF, it is left for future work to find a principled solution to account for missing covariance information in a computationally efficient manner.

# F  Error as a Function of Raw Computation Time

In addition to the previous anaylsis of the error in the approximate filtering estimate with varying low-rank dimensions, Figures F.1a and F.1b show the error as a function of wall-clock computation times for the linear-advection model (Section 4.1) and the spatio-temporal Matérn model (Section 4.3), respectively.

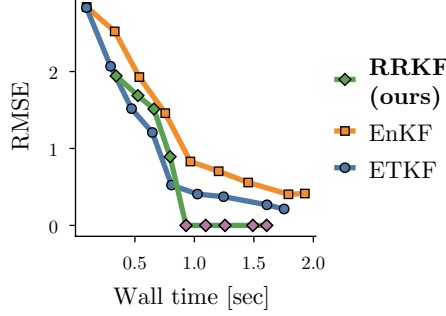

(a) *Linear advection dynamics (true rank = 51).*

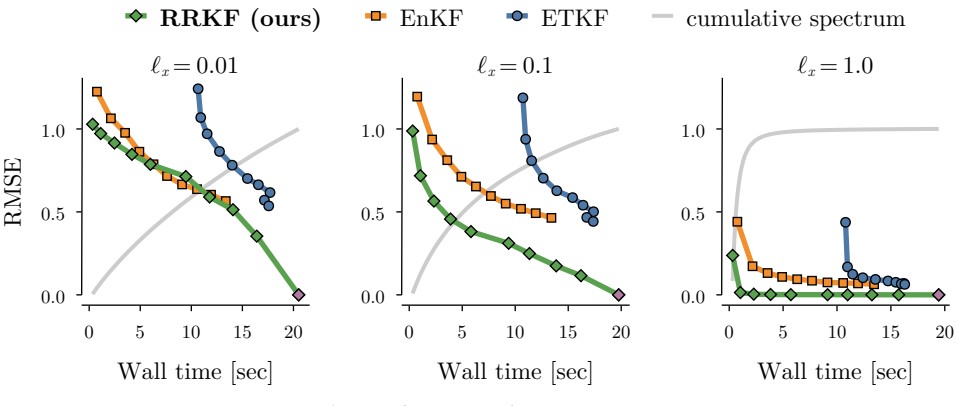

(b) *London air-quality regression.*

Figure F.1: *Error of the low-rank filters as a function of wall-clock computation time.* The raw computational expenses are comparable to those of the ensemble methods. The lower-left corner is the optimal setting with low error and low computation time. All methods approach this region of the plot for faster decays of the spectrum, while the RRKF performs better most of the time.