# OpenReview forum: "The Rank-Reduced Kalman Filter: Approximate Dynamical-Low-Rank Filtering In High Dimensions"
_NeurIPS.cc/2023/Conference — NeurIPS 2023 poster_

### Official Review · Reviewer_e5WD · 2023-06-26

**Soundness:** 3 good
**Presentation:** 2 fair
**Contribution:** 3 good
**Rating:** 7
**Confidence:** 4

**Summary:**

This paper develops a method for performing approximate Kalman filtering and smoothing using a low-rank approximation.  Fully low-rank predict, update and smoothing equations are derived.  Under certain reasonable assumptions, these equations reduce the cubic complexity to linear complexity (and quadratic if the assumptions are violated), while being “approximately exact” compared to Monte Carlo-based ensemble methods.  The method is evaluated on a range of problems and appears to perform favourably.  Sensible limitations and future opportunities are highlighted.


**Strengths:**

This paper addresses a critical issue – the super-linear (cubic) complexity of exact Kalman filtering/smoothing (KF/S).  This makes stable and closed-form estimation intractable for anything other than small problems.  The proposed solution is appealing: introduce a low-rank factorization that, through some linear algebra and recent solvers, is linear (or quadratic) in the rank.  While low-rank factorizations are common, I have not seen it applied to KF/S, a domain where it is very naturally appealing.

The detail in which the authors walk through the derivation is excellent, and appears to be mathematically sound (although I am not super familiar with this application domain).  The purely theoretical contributions of the paper are not huge; but are well-judged, interesting, apt for the problem at hand (modulo Limitations), and dovetail nicely with recent literature.

The experimental evaluation touches on a number of interesting examples that are high dimensional, and for which the proposed method seems to excel.  The final application, although a little off-the-wall, really does a good job in highlighting how high-dimensional this method can go.  Other experiments do a good job of comparing asymptotic to non-asymptotics in a range of domains.

I think there is definitely an audience at NeurIPS that will enjoy and really benefit from this work.  I also believe this work will engender follow-up work which will further build this direction.  The paper is on the whole well-written (modulo Weakness B) and is nicely visually presented.  Figures are very well prepared, with clearly legible fonts and clear colours.

**Weaknesses:**

The paper, on the whole, is very good, but I do think there are some areas for improvement.
- A:  The experimental evaluation is, on the whole, good.  Figure 1 is excellent in visually depicting how covariance matrices can be low rank.  WIth that in mind, I wonder if there is a simple, low-dimensional model that the authors can present that can be easily visualised.  I don’t have a great “feel” for any of the examples presented.  The results for this experiment certainly don’t need to be SoTA (since existing methods will probably also perform well).  Some visualisations (PCA projections of state, low-rank factors etc) can then be visualised to highlight the differences between methods and build reader intuition.  This example could be as simple as a synthetic LDS with dimension ~10.  Note:  I wouldn’t expect to see this during the rebuttal period, but I would _strongly_ encourage the authors to consider including it in a camera ready.
- B:  [I am aware this is a very general and somewhat hard-to-address comment, so please take these more as suggestions]  The paper itself is very notationally dense, to the point of obfuscating the simplicity and applicability of the method.  While the authors are to be commended for their attention to detail, I think parts of the paper can be thinned out, with more technical content being dropped to the supplement.  This will result in a paper that is more immediately digestible, and will have a larger reach and impact.  Some examples:
  -  I think Sec 3.1 is great for building reader intuition, but this intuition is then lost a little bit by having to wade through long math equations, where the details aren’t actually super important.
  - I  wonder if some of the smoothing equations can be cut to the supplement, with just the final forms retained and note directing the interested reader to the supplement.  This newly cleared space can then be used for including higher-level sketch proofs, intuition-building worked examples or text, algorithm blocks, diagrams, experiment visualisations etc.
  - Including a banner figure (and maybe even an algorithm block) explaining the whole method would be nice, just to visually complement the math.
  - Maybe consider including a table outlining the positives and negatives of each approach under particular operating conditions?  Then the benefits are stated clearly and concisely outside of the body text.
  - **The key takeaway** is that you have already solved the hard equations and derived an elegant solution!  Therefore, I would encourage you to strip out as much complexity from the main text as possible, and add extra emphasis to the end result – even something as cartoonish as a callout box with the key equations (akin to how Sarkka has a full equation block per model/inference methodology).

## Minor weaknesses:
- It would be nice if the DLRA/BUG integrator/solver were introduced in the main text.  You highlight it as a key component, but there is not really much discussion of its implementation, assumptions, limitations, complexity etc.
- I don't quite understand the $(\Phi \Sigma \quad Q)$ notation in (16) (specifically the white space between the matrices).  Please explicitly define it somewhere.

If these changes (and the limitations)  were made/commented on, then I would consider upgrading my review.


**Questions:**

Also see weaknesses and limitations.

Is operating in the continuous time domain strictly necessary?  My understanding is that a lot of complexity (at least notationally) goes away when you consider discrete time/equally spaced observations.  I think all of your experiments used equally spaced observations as well.  It could be simpler if you present the material in the discrete time domain, and then have a paragraph outlining that this also extends to the continuous time/SDE domain.


**Limitations:**

A limitation is that the authors do not consider bounding the error introduced by the low-rank approximation.  For instance, the LML in (12) is for the approximate model.  There is no comment on the approximation gap between the approximate model and the full-rank model.  I am not fully sure on this, but I think it is possible to construct a bound based on the truncated parts of the spectrum of the matrix being truncated.  If this isn’t possible, some qualitative discussion would help reinforce this point.  If you were able to construct a bound – any bound! – on the error, then that would put a nice bow on the approximation, as opposed to relying on evaluating the error empirically.

---

> ### Author Rebuttal · Authors · 2023-08-09
>
> Many thanks for your strong review and your detailed, positive assessment of our work! We greatly appreciate your appraisal that "there is definitely an audience at NeurIPS that will enjoy and really benefit from this work".
> In the following we will address your remaining concerns, which we hope further improves and solidifies your valuation of the quality of our work.
>
> **Q: Writing suggestions**
>
> A: Many thanks for taking the time to formulate concrete suggestions in order to help improve our work! We will consider them for the camera-ready version.
>
>
> **Q: Is there a simple, low-dimensional model that can be easily visualized, in order to give more intuition into the workings of the method?**
>
> A: We have given this idea some thought. It is not easy to construct an instructive example that is low-dimensional (and thus easily visualizable) and can be further compressed with a low-rank method. The full potential of our contribution unfolds exactly in very large systems in which the state-covariance can be tracked approximately on a low-dimensional manifold. Thus, we went in the opposite direction to focus on this by showcasing the method on a very large dynamical system with the example on rainfall prediction. There, we wanted to convey the intuition that a lot of locations have to be tracked but neighboring points will exhibit similar error estimates, which justifies the low-rank assumption.
>
>
> **Q: Is operating in the continuous time domain strictly necessary?**
>
> A: In many cases, continuous formulations of dynamics are sparse, which does not in general transfer to the discretized model. This would thus not allow for exploiting the sparse properties of the continuous system for computational tractability.
>
>
> **Q: Could a bound to the error of the method be provided?**
>
> A: This brings up an interesting point. However, it is not at present clear how to approach an error analysis in a tractable way. While there are initial ideas for _local_ error bounds, we saw value in postponing their systematic investigation to future work in order to place more focus on reader intuition behind the approach.
>
>
> **Q: It would be nice if the DLRA/BUG integrator/solver were introduced in the main text.**
>
> A: We give an intuition behind the method in Section 2.2 and the concrete algorithm as we use it in Section C of the supplementary material. We further provide all necessary references to the DLRA literature. In our opinion this is a good usage of the limited space for an existing method with a large body of existing literature.
>
>
> **Q: The notation for square-root factors of covariances (e.g. Eq. (16)) is somewhat unclear.**
>
> A: We will clarify the notation in a revised version of the paper. For example, $(\Phi \Sigma^{1/2} \quad Q^{1/2})$ denotes a block matrix that is a square-root factor of the predicted covariance matrix, in that the outer product $(\Phi \Sigma^{1/2} \quad Q^{1/2})(\Phi \Sigma^{1/2} \quad Q^{1/2})^\top = \Phi\Sigma\Phi^\top + Q$. If $\Sigma^{1/2}$ and $Q^{1/2}$ are $n\times r$ - matrices -- i.e. low-rank square-root factors of $\Sigma$ and $Q$, respectively -- then the block matrix is in turn a low-rank square-root factor of the predicted covariance matrix.

---

> > ### Comment · Reviewer_e5WD · 2023-08-10
> > **Staying Put.**
> >
> > Thank you to the authors for their response.
> >
> > After reading your response, and the response to the other reviewers, I will maintain my scores as they were.
> >
> > With that said, I am very interested to see what the other reviewers have to say with respect to comparisons to existing literature.  I am not overly familiar with this domain, so I defer to the other reviewers in that respect.  However, even if there is overlap with existing works, I believe the clear presentation of a usable method is still valuable enough for inclusion, so long as the links to this existing work are thoroughly explored in the main text.
> >
> > Good work, and good luck.
> >
> > e5WD

---

### Official Review · Reviewer_2SHM · 2023-07-02

**Soundness:** 3 good
**Presentation:** 3 good
**Contribution:** 3 good
**Rating:** 7
**Confidence:** 4

**Summary:**

The paper proposes an approximation method for Kalman filtering of intractable, high-dimensional Gauss-Markov settings with linear observations. It adopts existing DLRA methods to compute a discrete-time representation of the sampled process and suggests a recursive filtering and smoothing methods in a low-rank formulation.



**Strengths:**

To the best of my knowledge, the proposed filtering scheme is novel and is shown to outperform existing EnKFs under the same configurations. As opposed to EnKF approximation methods, this method is deterministic and capable of accurately restoring the optimal filter given the right parameterization.

Derivation of the recursive filter seems correct, and filtering is done in a square-root fashion, which is stable and efficient. The recursion step is shown to be efficient (although under very strong assumptions). The method additionally suggests an efficient calculation for the observation likelihood.

**Weaknesses:**

The method assumes the low rank $r$, which is a drawback of many low-rank approximation methods. Even if it is beyond the scope of the paper, there should be a short discussion about choosing this parameter or mention this as a limitation.

In Figs. 2, 3 the RRKF is compared with two other methods, both are Monte-Carlo, and are not expected to restore the exact filter. A righteous comparison should include some deterministic algorithms e.g. [10], [r1] below.

[r1] MLA	Farrell, Brian F., and Petros J. Ioannou. "State estimation using a reduced-order Kalman filter." Journal of the Atmospheric Sciences 58.23 (2001): 3666-3680




**Questions:**

1. The recursive methods assumes knowing $\Sigma_{l-1}$ at each step, but don't mention how to initialize $\Sigma_{1}$ at $t_1$. Is it just $Q_{t_1}$?

2. The estimations on Rainfall in Australia regression (Fig.5) don't seem very accurate, especially at spatial points where data deviates from the mean value. Authors claim that "the RRKF achieves high-quality approximate estimates", but I'm not convinced. Can they support this claim quantitively?

Minor comments:

Eqn. (4) and (16) are somewhat unclear. It would be helpful to clearly state matrices dimensions.

In l.146 it is stated that truncated SVD complexity is $O(nr^2)$, which seems to be right. But can you explain/reference this claim?


**Limitations:**

The method assumes the low rank $r$. This limitation is not discussed.

---

> ### Author Rebuttal · Authors · 2023-08-09
>
> Many thanks for your detailed and positive assessment of our work. In the following we will address your remaining concerns, which we hope further improves and solidifies your valuation of the quality of our work.
>
> **Q: The method introduces the low-rank dimension $r$ as a tunable parameter of the method. How would one choose $r$?**
>
> A: Thank you for raising this important point. It is true that the method introduces the low-rank dimension $r$ as a tunable parameter. We will emphasize and discuss this in our revision. In fact, the dynamic low-rank approximation (DLRA) literature offers a potentially quite interesting direction for follow-up work, which might automate the choice of $r$ [2].
>
>
> **Q: The method should be compared to reduced-order Kalman filters ([1]).**
>
> A: [1], like us, takes the model as given but heavily relies on realization theory of stable linear time-invariant systems to obtain a reduced-order model.
> In contrast, the dynamic low rank method that we employ can be applied to any matrix differential equation.
> Consequently, we expect our present method to be easier to extend to other model classes (i.e. nonlinear). Therefore, we view a comparison to the method presented in [1] as not critical to demonstrate the value of our method. Nevertheless, we will take the opportunity to more clearly situate the present contribution within the larger body of work in low-rank methods for estimation and model identification.
>
>
> **Q: How is the state covariance matrix initialized?**
>
> A: In our experiments, we compute the stationary mean and a low-rank factorization of the stationary covariance matrix of the prior and condition the stationary moments on the first measurement in the respective time-series dataset. We will add this detail to the revised version.
>
>
> **Q: The rainfall predictions in the final experiment could be improved**
>
> A: You are right in noticing that the rainfall model yields less accurate predictions, as opposed to solving a PDE. Essentially, the rainfall data is interpolated approximately with a simple Matérn Gaussian-process model. This, of course, makes the prediction significantly cheaper, as well.
> Most importantly, the focus of this experiment is the high dimensionality of the state-estimation problem, which is solved efficiently and -- given the simple model and the significant rank compression -- accurately. We will clarify the focus of the experiment accordingly in the paper.
>
>
> **Q: The notation for square-root factors of covariances (e.g. Eq. (4) and (16)) is somewhat unclear.**
>
> A: We will clarify the notation in a revised version of the paper. For example, $(\Phi \Sigma^{1/2} \quad Q^{1/2})$ denotes a block matrix that is a square-root factor of the predicted covariance matrix, in that the outer product $(\Phi \Sigma^{1/2} \quad Q^{1/2})(\Phi \Sigma^{1/2} \quad Q^{1/2})^\top = \Phi\Sigma\Phi^\top + Q$. If $\Sigma^{1/2}$ and $Q^{1/2}$ are $n\times r$ - matrices -- i.e. low-rank square-root factors of $\Sigma$ and $Q$, respectively -- then the block matrix is in turn a low-rank square-root factor of the predicted covariance matrix.
>
>
> **Q: Why is the truncated SVD complexity $O(nr^2)$?**
>
> A: On tall and wide matrices the rank is upper-bounded by the smaller of both dimensions. It is therefore sufficient to compute only a subset ($r$) of the singular vectors of such a matrix, since the remaining singular values are zero. This is sometimes referred to as "compact SVD" (or "reduced SVD" (RSVD), or "economic SVD").
> This is detailed, e.g. in Chapter 5 in [3]. We will add this reference to the camera-ready version.
>
>
> [1] B. F. Farrell and P. J. Ioannou, "State estimation using a reduced order Kalman filter," 2001.
>
> [2] G. Ceruti, J. Kusch, and C. Lubich, “A rank-adaptive robust integrator for dynamical low-rank approximation,” 2022.
>
> [3] Golub, Gene H., and Charles F. Van Loan. Matrix computations. JHU press, 2013.

---

> > ### Comment · Reviewer_2SHM · 2023-08-13
> >
> > Thank you for your respose.

---

### Official Review · Reviewer_U69V · 2023-07-08

**Soundness:** 3 good
**Presentation:** 4 excellent
**Contribution:** 3 good
**Rating:** 6
**Confidence:** 4

**Summary:**

The paper presents a deterministic low-rank approximation of the Kalman filter for estimation in high-dimensional linear Gaussian setting. This setting is important for many problems that arise in meteorology and solved by Ensemble Kalman Filter (EnKF). The paper presents formula for low rank approximation of the covariance for both prediction and correction steps and demonstrates the proposed approach in different scenarios in comparison with EnKF.

**Strengths:**

- paper is well-written, has clear message
- it considers an important problem in meteorology
- the proposed approach is motivated well
- comprehensive numerical experiments are given

**Weaknesses:**

- the main weakness is the fact that in most high-dimensional data assimilation problems, one has has only access to a simulator for the forward dynamic update. It is not clear how the proposed approach perform low rank approximation in that case. In EnKF, one simulates each member of the ensemble according to the dynamic model
- the numerical code for reproducing the results are not given
- the relevance to NeurIPS audience is weak.



**Questions:**

1- What is a reference for stochasticity in EnKF introduces unfavorable properties. It will be good to be more concrete here, and explain what are exactly those unfavorable properties, since it is a central motivation for your approach.

2- The computational time plot in figure 4 is a bit surprising. How is that low rank approximation has smaller computational time compared to EnKF?

3- Can you produce a figure that demonstrates error as a function of raw computational time?

4- What is the reference for O(nr^2) complexity of truncated SVD?

**Limitations:**

yes

---

> ### Author Rebuttal · Authors · 2023-08-09
>
> Many thanks for your detailed and positive assessment! We hope that we can additionally resolve your main point of criticism in the following and thereby convince you to raise your score.
>
> **Q: It is not clear how to apply this method in case one has to forward the dynamics with a numerical simulator.**
>
> A: This is a very important point, the relevance of which we are very aware of. Since dynamics, which require a numerical simulator for forward propagation, are usually nonlinear, this question will receive systematic treatment in follow-up work, while this paper establishes the foundations of the method on a purely linear setup.
> Note that a simple, linear example of this case is showcased by the linear-advection example (Section 4.1; Fig. 1), which is a common baseline in papers that propose ensemble Kalman filter methods (e.g. [1], [2]). There, the simulation step is a simple matrix-vector product with a circulant (shift) matrix. Instead of individual ensemble members, the mean is simulated forwards and the linear operator is used to forecast the low-rank covariance matrix (as described).
>
>
> **Q: The code for reproducing the results is not given**
>
> A: The code is ready to be made public and will be released upon acceptance of the paper.
>
>
> **Q: The relevance to NeurIPS audience is weak.**
>
> A: We strongly disagree with this statement and hope to convince you of the opposite by addressing it here. The recent momentum in the domain of ML in the sciences, like, for instance, meteorology, geophysics, oceanography, etc., calls for efficient, yet accurate tools for the analysis of high-dimensional time series. Especially in sparse-data regimes, Gaussian-process models and hybrid physics-informed/data-driven ideas from the field of "data assimilation" become more and more attractive and often are preferred over purely data-driven approaches. Our method (and planned follow-up work) aims to serve as a well-founded, simple-to-use drop-in replacement for approximate Bayesian filtering/smoothing for exactly such applications and we hope to gain momentum in both the ML community and the sciences.
>
>
>
> **Q: What do you mean by "unfavorable properties" that are introduced by the stochasticity of the EnKF?**
>
> A: Adding sampled noise to the ensemble introduces sampling error [3,5], which leads to erroneous estimates of the approximate moments and motivates works like [3,1] and generally all work on deterministic ensemble Kalman filters; and it causes spurious correlations in the covariance estimator, which has then to be mitigated by localization techniques (e.g. [4]). The main point is that it is preferrable to approximately target the Karhunen-Loève truncation rather than sampling a random subspace.
>
>
> **Q: How is the RRKF faster than the EnKF in Fig. 4?**
>
> A: It is always difficult to compare absolute runtimes, due to nuances in the respective implementation. Even though we implemented all the methods ourselves and took great care in optimizing each of them to the same degree, there are likely variations. The point of our runtime analysis lies in the **asymptotic behavior**. This plot is not to show that our algorithm is faster than the EnKF. It is to show that the asymptotic runtimes of the ensemble-based algorithms is preserved by our method, while it has some major advantages over the stochastic counterparts, as described in the paper.
>
>
> **Q: Can you produce a figure that demonstrates error as a function of raw computational time?**
>
> A: Thanks for this suggestion. We included a draft of such figures in the PDF file attached to the general author rebuttal above. We would like to add those to the supplementary material of the camera-ready version.
>
>
> **Q: Why is the truncated SVD complexity $O(nr^2)$?**
>
> A: On tall and wide matrices the rank is upper-bounded by the smaller of both dimensions. It is therefore sufficient to compute only a subset ($r$) of the singular vectors of such a matrix, since the remaining singular values are zero. This is sometimes referred to as "compact SVD" (or "reduced SVD" (RSVD), or "economic SVD").
> This is detailed, e.g. in Chapter 5 in [6]. We will add this reference to the camera-ready version.
>
>
> [1] P. Sakov and P. R. Oke, “A deterministic formulation of the ensemble Kalman filter: an alternative to ensemble square root filters,” _Tellus A: Dynamic Meteorology and Oceanography_, vol. 60, no. 2, p. 361, Jan. 2008.
>
> [2] G. Evensen, F. C. Vossepoel, and P. J. van Leeuwen, _Data Assimilation Fundamentals: A Unified Formulation of the State and Parameter Estimation Problem_. in Springer Textbooks in Earth Sciences, Geography and Environment, 2022.
>
> [3] J. S. Whitaker and T. M. Hamill, “Ensemble Data Assimilation without Perturbed Observations,” _Mon. Wea. Rev._, vol. 130, no. 7, pp. 1913–1924, Jul. 2002.
>
> [4] A. Carrassi, M. Bocquet, L. Bertino, and G. Evensen, “Data assimilation in the geosciences: An overview of methods, issues, and perspectives,” _WIREs Clim Change_, vol. 9, no. 5, Sep. 2018.
>
> [5] W. Sacher and P. Bartello, “Sampling Errors in Ensemble Kalman Filtering. Part I: Theory,” _Monthly Weather Review_, vol. 136, no. 8, pp. 3035–3049, Aug. 2008.
>
> [6] Golub, Gene H., and Charles F. Van Loan. Matrix computations. JHU press, 2013.

---

> > ### Comment · Reviewer_U69V · 2023-08-15
> >
> > Thanks for the detailed response my questions and providing clarifications. My review stays positive, but I can not increase because  I still find the model a major practical limitation that underlies the contribution. And I find the run-time results surprising as ETKF and EnKF have the almost the same curve, although ETKF involves solving a linear program scaling with the number of particles.

---

> > > ### Author Response · Authors · 2023-08-15
> > >
> > > Many thanks for continuing to argue in favor of acceptance and for detailing your remaining concerns!
> > > We would like to take this opportunity to address in particular your second concern briefly, in the hope that it can be satisfactorily resolved in the course of a brief discussion.
> > >
> > > It is correct that the ETKF scales cubically in the number of particles $r$, which does not show in the experiment on asymptotic runtime. However, in this experiment, we investigated the asymptotic runtime with respect to the _state-dimension_ $n$ and let $r$, the number of particles, be fixed at a very small number $r = 5$, in order to take this quantity out of the analysis. This experiment was to show that the proposed method - as the ETKF - has the important quality that its asymptotic complexity does _not_ scale cubically in a typically very large number ($n$).
> > >
> > > As for your other concern, we agree with the assessment that the assumed linear state-space model is a remaining limitation of the approach, as addressed in the limitations section. We continue to argue that this extension is best followed-up upon in future work after the foundations of the method have been established in the present contribution, on a linear-dynamics setup.
> > >
> > > We would be very pleased to continue the discussion if there are open questions remaining and thank you again for your follow-up comment.

---

### Official Review · Reviewer_HH4H · 2023-07-25

**Soundness:** 2 fair
**Presentation:** 3 good
**Contribution:** 2 fair
**Rating:** 4
**Confidence:** 3

**Summary:**

The paper proposes a deterministic low rank approximation to the Kalman filter in high dimensional settings where the computational complexity of the full Kalman filter is very expensive. They compare to the ensemble Kalman filter and ensemble transform Kalman filter.

**Strengths:**

The paper is well written.

**Weaknesses:**

See questions. The biggest issue is that the comparisons are not against “fair” competing methods (any low rank Kalman filter). They also do not mention anything about the large swaths of low rank variants of Kalman filters.

**Questions:**

The scenario for this problem is linear dynamic systems? Ensemble style Kalman filters are often used in non-linear dynamic systems and have the greatest advantage there; including computational ones, but that is compared to particle filters etc. . The authors cite using ensemble Kalman filters for computational cheapness, but the vast majority of those papers are from not very good journals and their accuracy is suspect...

In contrast the real data experiments are for atmospheric problems that generally have non-linear dynamics?

How are you computing the low rank dimension of the ensemble Kalman filter, which is not a low rank method (figure 2)? Neither the ensemble Kalman filter nor the ensemble transform Kalman filter are explicitly attempting to estimate a low rank approximation. It would be more appropriate to compare against a variant of the Kalman filter that is a low rank approximation (or a sparse one at least). There are tons of variants from just a quick google search.


**Limitations:**

Yes

---

> ### Author Rebuttal · Authors · 2023-08-09
>
> Many thanks for your critical assessment of our work!
> We identified your main points of contention to be (a) that the ensemble Kalman filter is not a low-rank method and (b) we are not comparing our method to a low-rank filter. Both of these points are based on a misconception, which we will explain below, followed by addressing your remaining concerns. We believe that our response demonstrates that the points raised do not warrant a rejection of our work.
>
> **Q: The ensemble Kalman filter is "not a low-rank method" and thus not a fair comparison.**
>
> A: This statement is incorrect. The EnKF *is* indeed a low-rank method, where the rank of the covariance approximation is determined by the number of ensemble members. Concretely, $\Sigma \approx (\frac{1}{\sqrt{r-1}}X)(\frac{1}{\sqrt{r-1}}X)^\top$, where the ensemble $X \in \mathbb{R}^{n \times r}$ is of rank $r$ and $r \ll n$. When comparing our method to the EnKF in our experiments, the "low-rank dimension" of the EnKF thus corresponds to the ensemble size.
> In view of this, we would agree that our experimental comparisons are not exhaustive.
> However, we do maintain that our experimental results are certainly sufficient to demonstrate the merits of our approach.
> Furthermore, on the basis of this review and that of dWp4 we are confident that we can more clearly situate the present contribution within the larger body of work in low-rank methods for estimation and model identification.
>
> **Q: The problem statement contains only linear systems, whereas the EnKF is often used for nonlinear systems.**
>
> A: This paper is the first in what we hope to be a line of work. The method is motivated by being entirely deterministic -- which clearly has benefits over Monte-Carlo-based approaches in some relevant scenarios -- and it is established here on linear systems. Indeed our algorithm, as presented in the paper, only applies to linear systems. As you are aware, we explicitly discuss this in the limitations section of our work. Indeed, many state-estimation problems are nonlinear, and this setting will certainly be followed-up upon in future work. The experiment on rainfall estimation succeeds in showcasing the computational feasibility in a high-dimensional state-estimation problem; the assumption of linear dynamics does not impede this argument.
>
>
> **Q: Cited literature that uses ensemble Kalman filters for computational cheapness is questionable**
>
> A: We fundamentally reject this criticism. The foundational paper by Evensen [1], establishing the first version of the EnKF, presents the method as "a better alternative than solving the traditional and computationally extremely demanding approximate error covariance equation used in the extended Kalman filter". Our method solves precisely this problem.
> Further, in an important foundational piece on the EnKF [2], Houtekamer and Mitchell write:
>
> > "In particular, to make the ensemble approximation feasible, we have to use a fairly small ensemble with many less members than either the number of model coordinates, or the number of independent observations, or the (unknown) dimension of the dynamical system. To nevertheless obtain good results, we must (i) counter the tendency of the ensemble spread to underestimate the true error, and (ii) localize the ensemble covariances. The localization is severe and leads to imbalance in the initial conditions."
>
> This clearly states the agenda of developing a computationally tractable approximate inference scheme and the need to counteract degeneracies that come with the naïve ensemble-based approximation.
>
> [1] G. Evensen, (1994), Sequential data assimilation with a nonlinear quasi-geostrophic model using Monte Carlo methods to forecast error statistics, _J. Geophys. Res._,  99(C5),  10143–10162.
>
> [2] P. L. Houtekamer and H. L. Mitchell, “Ensemble Kalman filtering,” Q. J. R. Meteorol. Soc., vol. 131, no. 613, pp. 3269–3289, Oct. 2005.
>
> **Q: Much of the existing low-rank Kalman filters are not mentioned.**
>
> A: We will extend the related work section to position the present contribution in the broader context of dimensionality reduction in high-dimensional state-estimation problems.

---

> > ### Comment · Reviewer_HH4H · 2023-08-19
> > **Response to Rebuttal**
> >
> > Thanks for the response. While the ensemble Kalman filter indeed produces a low-rank covariance where the rank is the size of the ensemble size, it is not purposefully estimating a "good" low rank covariance. The ensemble Kalman filter produces a low rank matrix in the same fashion as the sample covariance matrix is a "low rank matrix" when there are fewer samples than dimensions. But just as the SCM is not a "good" estimator of the true covariance matrix (and is not even positive definite, which violates the required properties of covariance matrices), the standard ensemble Kalman filter is not producing a "good" estimator of whatever is the true prior covariance. There are numerous other methods that assume that the hidden states of the dynamics systems lie on a lower dimensional manifold and explicitly attempt to estimate good low rank estimator for the covariance. The current comparison to the ensemble Kalman filters is like comparing against an explicitly estimated low rank covariance estimator (there are a lot of methods that do this e.g. with regularization) to a baseline sample covariance matrix that has insufficient samples. That is our issue with comparing to the ensemble Kalman filter.
> >
> > In respond to the quote. Yes using fewer ensemble members makes the process computationally cheaper, but at what cost? The original quote downplays the cost of this "computational cheapness". Personally I find the original argument extremely weak (this is not the fault of the authors, but choosing to use it is). Normally approximate methods / optimization relations / etc. always compare against the original solution to show the cost of the cheaper computations. Otherwise a similar argument could be that using fewer samples / data points makes any method computational cheaper...

---

> > > ### Author Response · Authors · 2023-08-21
> > > **An attempt to clear up remaining misconceptions**
> > >
> > > We would like to thank the reviewer for engaging in further discussion.
> > > However, we find it strange that the reviewers main objection appears to be based on their dislike of the ensemble Kalman filter,
> > > rather than an assessment of the quality of our proposed alternative.
> > > We further note some remaining misconception. We elaborate on these issues below.
> > >
> > > **On the ensemble Kalman filter**
> > >
> > > It is indeed the case that the ensemble Kalman filter leaves a lot of room for improvement in terms of tracking a low rank approximation to the state covariance matrix.
> > > In fact, the situation is even worse as ad-hoc tricks to improve the approximation quality often needs to be employed in practice.
> > > Nevertheless, the ensemble Kalman filter is often used practice, which is precisely why we find it pressing to address these deficiencies in algorithm design.
> > >
> > > The reviewer is entitled to their opinions on the arguments adduced in the literature on ensemble Kalman filter.
> > > However, it is rather strange to take our citation of the arguments for developing the ensemble Kalman filter as endorsement of the same.
> > > The purpose of the citation was to demonstrate that, contrary to the reviwers belief, a primary reason behind the development of the ensemble Kalman filter was indeed to reduce computational cost - this is correct.
> > >
> > > **On comparisons against the original solution**
> > >
> > > The reviewer suggests that approximate methods should be compared against the original solution to show the cost (in terms of accuracy) paid for the lessened cost in computation.
> > > We completely agree! In fact we have done this in the original manuscript, the reviewer is invited to examine Figures 2 and 3 to see the result.
> > >
> > >
> > > **On alternative methods**
> > >
> > > The reviewer mentions "numerous other methods" for attacking the present problem formulation.
> > > However, not a single specific example is given.
> > > We would of course be happy to appropriately address specific suggestions for related work if examples were given, see e.g. the responses to dWp4 and 2SHM.
> > >
> > > **On positive definiteness of covariance matrices**
> > >
> > > Lastly, we would like to do away with the misconception that covariance matrices need to be positive definite,
> > > indeed it is sufficient that they are positive semi-definite.
> > > This property (positive semi-definiteness) of course holds for the sample covariance matrix, the ensemble Kalman filter approximation, and our reduced rank method.

---

> > > > ### Comment · Reviewer_HH4H · 2023-08-21
> > > >
> > > > My point is not some "dislike" of the ensemble Kalman filter. It is that this is not a "fair" comparison to compare against. While the ensemble Kalman filter can be used as a baseline, it is far too simple and you should also show your proposed method works against more recent, more complex methods that are also improvements to the baseline ensemble Kalman filter and attempt to explicitly estimate a low rank covariance matrix. For example, in your response to dWp4 you mention that
> > > >
> > > > "[4], like us, takes the model as given but heavily relies on realization theory of stable linear time-invariant systems to obtain a reduced-order model. In contrast, the dynamic low rank method that we employ can be applied to any matrix differential equation. Consequently, we expect our present method to be easier to extend to other model classes (i.e. nonlinear)"
> > > >
> > > > This may be the case that your method is easier to extend to other model classes, but your experiments are for linear systems. It would be a much more "fair" comparison to show the performance of your method against a model such as this one. My previous comment was not about how I "dislike" ensemble Kalman filters, but how unfair of a comparison it was to only compare against such a simple model. I am not the only reviewer to point out that you should be comparing to other low rank methods. Reviewer 2SHM has done the same (perhaps more elegantly than myself).

---

### Official Review · Reviewer_dWp4 · 2023-07-27

**Soundness:** 3 good
**Presentation:** 2 fair
**Contribution:** 2 fair
**Rating:** 5
**Confidence:** 4

**Summary:**

Filtering (estimation) and smoothing of large-dimensional state-space models are computationally challenging. Exact Kalman filtering, at least insofar as LTI SDEs are considered, is a mature solution for such tasks but the cubic complexity of such methods makes it computationally intractable. This paper proposes an approximate Gaussian filtering and smoothing method that propagates low-rank approximations of the covariance matrices. To enable this, i) Lyapunov equations are projected into a manifold of lower rank in the prediction step combined with ii) square root filtering, which together offer numerically stable and tractable solution. Four examples are provided, with increasing complexity, and in one case runtime analysis is also carried out to support the claims. Two measures are used RMSE and covariance deviation to assess the method’s performance. Corollary 1 provides a “Kalman-like” result (an important contribution of the paper) while Section 3.2 discusses the algorithm of the method for filtering (3.4 for smoothing) and section 3.3 provides the time complexity of the method.

**Strengths:**

The proposed method differs from existing ensemble-based approaches in that the low-rank approximations of the covariance matrices are deterministic, rather than stochastic. This is important because it allows the method to reproduce the exact Kalman filter as the low-rank dimension approaches the true dimensionality of the problem.

The time complexity analysis is well-written, and authors clearly demonstrate their method reduces computational complexity from cubic (for the Kalman filter) to quadratic in the state-space size at most (and linear in some cases—refer to proposition 3 and assumptions therein).

**Weaknesses:**

My major issue with this paper is there is a large body of work on estimation of high- and infinite-dimensional dynamical systems and this paper does not position itself in such context. There are numerous works on using sparse identification (Sparse reduced-order modeling: Sensor-based dynamics to full-state estimation), Koopman-based (A Robust Data-Driven Koopman Kalman Filter for Power Systems Dynamic State Estimation), balanced-truncation ROMs and Kalman (State Estimation Using a Reduced-Order Kalman Filter), DMD-based ROM and Kalman (Dynamic mode decomposition and robust estimation: Case study of a 2d turbulent boussinesq flow), etc. In above-mentioned works, usually the high-dimensional governing equations is first projected in some appropriate basis (low-rank), and then an appropriate filter is deigned. Once estimation is done, the reconstruction is projected back into the physical domain. This also enables the application of the method to i) nonlinear dynamical system, ii) unknown systems. Such approaches benefit from lower computational cost, similar to the proposed method in the paper. In a sense authors acknowledge, more or less, such line of work in the last paragraph of Section 1. However, when they write “In contrast to our work, their method assumes that the dynamics unfold entirely in a lower-dimensional space, and conditioning on measurements happens by transporting the low-dimensional diffusive process to the full space by a projection that is also assumed to be known” there is no quantitative or qualitative comparison to support their claim. Let’s say for example 1 considered in this paper, authors first project the PDE into low rank ODE, and then design the standard Kalman filter in such a manifold, and once estimation is propagated in time, the low dimension is converted back into actual space. What are the advantages and disadvantages? Please note I am not simply asking for more comparison (in addition to, say, EnKF) but more a conceptual question of what the real advantage of the proposed method is to state of the art for ROM-based estimation. I’m sure authors are aware of works like “Krylov subspace methods for solving large Lyapunov equations” where large scale Lyapunov equations are solved with efficient algorithms. I hope authors clarify their novelties in the rebuttal.

**Questions:**

In section 1, Please be more specific about such unfavorable properties (line 45). Overall, the issue of GP is not discussed clearly in this section and elsewhere in the manuscript.

Any insights if such approach can be applied to nonlinear systems? What if the dynamics is unknown (A and B)?

I'd suggest moving section 4.4 as either last section or right after 4.1. Currently the flow of the results section is hard to follow: first three examples are given to demonstrate the accuracy of the model with the given two measures, then a discussion of runtime and finally again another result for rainfall data.

It would be helpful to include a discussion on the impact of snapshots noise (measurement noise) on the results.

---

> ### Author Rebuttal · Authors · 2023-08-09
>
> Many thanks for your careful reading of our paper as well as your thoughtful critique.
> In our response, we would like to divide the major points of contention into two categories: "Address literature on reduced order methods" and "model order reduction versus our approach".
> These will be responded to in order, followed by answers to more specific questions raised.
> It is our assessment that an elaboration on related work in the camera-ready would be sufficient to address your concerns,
> and it is our hope you come to the same conclusion upon reading our response.
>
> Before proceeding we shall refer to the papers cited in your review as follows
>
> - [1] Sparse reduced-order modeling: Sensor-based dynamics to full-state estimation
> - [2] A Robust Data-Driven Koopman Kalman Filter for Power Systems Dynamic State Estimation
> - [3] Dynamic mode decomposition and robust estimation: Case study of a 2d turbulent boussinesq flow
> - [4] State Estimation Using a Reduced-Order Kalman Filter
> - [5] Krylov subspace methods for solving large Lyapunov equations
>
> and we will also refer to some of the references of our paper by
>
> - [6] P. Sakov and P. R. Oke, “A deterministic formulation of the ensemble Kalman filter: an alternative to ensemble square root filters,” 2008.
> - [7] J. S. Whitaker and T. M. Hamill, “Ensemble Data Assimilation without Perturbed Observations,” 2002.
> - [8] A. Carrassi, M. Bocquet, L. Bertino, and G. Evensen, “Data assimilation in the geosciences: An overview of methods, issues, and perspectives,” 2018.
> - [9] W. Sacher and P. Bartello, “Sampling Errors in Ensemble Kalman Filtering. Part I: Theory,” 2008.
>
> ### Address literature on reduced order methods
>
> The cited papers [1,2,3] are concerned with low-rank modelling of dynamical systems from measured data.
> The paper [4] is concerned with low-rank approximations of a given model based on classical theory on linear time invariant systems.
> Once a low rank model is obtained for the phenomena under study then state estimation becomes computationally trivial, in the sense that the model is not very large.
> These contributions are of course related to the present paper under the broader context of low-rank/ dimensionality reduction methods.
>
> However, they do differ on some key points:
>
> - [1,2,3] allow themselves to construct a (low-rank) model from measured data. Whereas we assume the model class is fixed and the objective is make inference tractable with minimal violence done to the model.
> - [4], like us, takes the model as given but heavily relies on realization theory of stable linear time-invariant systems to obtain a reduced-order model.
> In contrast, the dynamic low rank method that we employ can be applied to any matrix _differential_ equation.
> Consequently, we expect our present method to be easier to extend to other model classes (i.e. nonlinear).
>
> Finally, we will touch on the mention of paper [5], which develops methods for obtaining low-rank approximations to the solutions of _algebraic_ Lyapunov equations.
> Whereas the problem we tackle with the dynamic low rank method is to obtain low-rank approximations to the solution of Lyapunov _differential_ equations.
> Consequently, the methods of [5] can not directly be brought to bear on our problem.
>
> ### Model order reduction versus our approach
>
> The review brings up example 1 in the paper to support a discussion of model order reduction (i.e. [1,2,3,4]) versus to the present approach
> of simply tracking a low rank approximation to the covariance matrix while keeping the model intact.
>
> Broadly speaking, the approaches of [1,2,3,4] are expected to work well under the following assumption:
>
> - [A1] The state evolves in a low-dimensional manifold (approximately).
>
> Whereas our approach is expected to work well under the following assumption:
>
> - [A2] the error in the state estimate evolves in a low-dimensional manifold (approximately).
>
> These two assumptions are quite different indeed and of course come with certain advantages and disadvantages.
> We can not say how the quality of state estimation would differ between the two approaches under [A1],
> we do however note that the model reduction approach would be more economical, certainly in storage, but perhaps also in computation.
> On the other hand, under [A2] it is not clear that the model reduction approach would be appropriate in all cases.
> In particular it could be the case that the state of the system really does develop in a high-dimensional space whereas the uncertainties are concentrated in a lower dimensional subspace
> -- in this case we expect our proposed approach to be more succesful.
>
> **Q: What is meant by "unfavorable properties" of the stochastic nature of the EnKF?**
>
> A: Adding sampled noise to the ensemble introduces sampling error [7,9], which leads to erroneous estimates of the approximate moments and motivates works like [7,6] and more work on deterministic ensemble Kalman filters. It further causes spurious correlations in the covariance estimator, which has to be mitigated, e.g., using localization techniques (e.g. [8]). The main point is that it is preferrable to approximately target the Karhunen-Loève truncation rather than sampling a random subspace.
>
> **Q: Your work only regards linear dynamics systems, whereas existing methods are usually applied to nonlinear dynamics**
>
> A: Indeed our algorithm, as presented in the paper, only applies to linear systems. As you are aware, we explicitly discuss this in the limitations section of our work. Indeed, many state-estimation problems are nonlinear, and this setting will certainly be followed-up upon in future work.
>
>
> **Q: Writing suggestions**
>
> A: Many thanks for taking the time to formulate concrete suggestions in order to help improve our work! We will consider those for the camera-ready version.

---

> > ### Comment · Reviewer_dWp4 · 2023-08-15
> >
> > Thanks authors for their detailed response. I am in the process of reading them carefully.

---

### Author Rebuttal · Authors · 2023-08-09

We are very grateful to all reviewers for their high-quality reviews and for providing constructive feedback. We are inspired by the reviewers' confirmation that our work addresses a "critical issue" (e5WD) and an "important problem" (U69V) to which our "proposed solution is appealing" (e5WD).
We would like to thank the reviewers for judging our manuscript as "nicely visually presented" (e5WD), "well-written" (e5WD, U69V, HH4H, dWp4), conveying a "clear message" (U69V); and for highlighting the novelty of the method (e5WD,2SHM) and the "excellent" (e5WD) and "mathematically sound" (e5WD) derivation. We are particularly encouraged by the assessment that "there is definitely an audience at NeurIPS that will enjoy and really benefit from this work" (e5WD).

Further, we are grateful to the reviewers for suggesting improvements, which will greatly benefit a revised version of the manuscript.
The most significant objection is that the relation of the proposed method to other existing approaches to low-rank filtering should be made more explicit. We will take the opportunity to situate the present contribution within the larger body of work in low-rank methods for estimation and model identification in the camera-ready version.

---

### Decision · Program_Chairs · 2023-09-21

**Decision:**

Accept (poster)

**Comment:**

The paper presents a new low-rank approximation method for Gaussian filtering and smoothing in high-dimensional dynamical systems. It significantly departs from previous ensemble-based methods.

The authors develop a time complexity analysis that is well-detailed and compelling.Addressing the issue of super-linear complexity in exact Kalman filtering and smoothing, the paper proposes a solution using low-rank factorization. This solution reduces complexity to linear or quadratic levels, making it effective for larger problems.

Although the paper doesn't provide major theoretical contributions, it addresses a crucial challenge and aligns with the most recent literature. The paper's strength lies in its practical application demonstrated through a variety of experiments. This validates the proposed method effectively. Feedback suggests further improvements, such as the inclusion of specific baselines, and the authors should do so in the camera-ready version. Based on these observations, the paper is recommended for acceptance.